# Primary, Secondary, and Tertiary Prevention of Congenital Cytomegalovirus Infection

**DOI:** 10.3390/v15040819

**Published:** 2023-03-23

**Authors:** Pauline Sartori, Charles Egloff, Najeh Hcini, Christelle Vauloup Fellous, Claire Périllaud-Dubois, Olivier Picone, Léo Pomar

**Affiliations:** 1School of Health Sciences (HESAV), University of Applied Sciences and Arts Western Switzerland, 1011 Lausanne, Switzerland; 2Department Woman-Mother-Child, Lausanne University Hospital and University of Lausanne, 1011 Lausanne, Switzerland; 3Assistance Publique-Hôpitaux de Paris APHP, Nord, Service de Gynécologie Obstétrique, Hôpital Louis Mourier, 92700 Colombes, France; 4Université de Paris, 75006 Paris, France; 5INSERM, IAME, B.P. 416, 75870 Paris, France; 6Department of Obstetrics and Gynaecology, West French Guiana Hospital Center, French 97320, Guyana; 7CIC Inserm 1424 et DFR Santé Université Guyane, 97320 ST Laurent du Maroni, France; 8Université Paris-Saclay, INSERM U1193, 94804 Villejuif, France; 9Laboratoire de Virologie, AP-HP, Hôpital Paul-Brousse, 94804 Villejuif, France; 10Groupe de Recherche sur les Infections Pendant la Grossesse (GRIG), 75000 Paris, France; 11Virology Laboratory, AP-HP, Sorbonne Université, Hôpital Saint-Antoine, F-75012 Paris, France

**Keywords:** cytomegalovirus, congenital infection, prenatal diagnosis, prevention

## Abstract

Cytomegalovirus infection is the most common congenital infection, affecting about 1% of births worldwide. Several primary, secondary, and tertiary prevention strategies are already available during the prenatal period to help mitigate the immediate and long-term consequences of this infection. In this review, we aim to present and assess the efficacy of these strategies, including educating pregnant women and women of childbearing age on their knowledge of hygiene measures, development of vaccines, screening for cytomegalovirus infection during pregnancy (systematic versus targeted), prenatal diagnosis and prognostic assessments, and preventive and curative treatments in utero.

## 1. Introduction

Prevention and early detection of infection are an integral part of most antenatal care programs recommended by health policy makers to ensure effective prevention strategies against mother-to-child transmission of pathogens [1]. The management of infections during pregnancy represents a medical challenge for practitioners who not only must integrate an infected woman but also the developing fetus into a therapeutic program [2]. Cytomegalovirus (CMV) is a DNA herpes virus within the family of β-herpesviruses (HHV-5: human herpesvirus 5) that is found worldwide with a variable prevalence, according to geographical regions and socio-economic status, and with an estimated overall seroprevalence of 83% [3,4]. A recently published systematic review estimated the global seroprevalence of CMV to be 86% in women of childbearing age [4,5]. Its seroprevalence tends to be higher in lower socio-economic groups, racial and ethnic minority populations, and women of high parity and advanced maternal age [5]. CMV is also known as the most common infectious agent causing teratogenic congenital infection [6,7]. Affecting an average of 2.3% of pregnant women and about 0.5 to 1% of live births in Europe each year [6,7,8,9], CMV has, thus, become a major public health problem in many countries [4]. Congenital CMV (cCMV) is the leading cause of non-genetic neurosensory impairment in children, responsible for 10% of all cases of cerebral palsy among them [2,10,11,12]. Although most infected newborns (85–90%) are considered asymptomatic at birth, cCMV is associated with a higher risk of hearing loss and late-onset neurodevelopmental disorders (in 5–15% of “asymptomatic” newborns) [13]. To reduce the likelihood of maternal infection, maternal–fetal transmission, and subsequent complications in the event of a congenital infection, prevention strategy is structured around three key components: primary, secondary, and tertiary prevention [14,15,16,17]. In 2012, the World Health Organization (WHO) published a list of public health operations to help countries optimize and monitor the health and well-being of the population within their jurisdiction. Disease prevention has a prominent place in this list and includes these three main axes (EPHO 5). According to the WHO, the implementation of these activities within a health system is the prerequisite for its performance [18]. Thus, the aim of this review is to present the different strategies available in the prevention of cCMV and to assess their efficacy.

## 2. Primary Prevention: To Reduce the Risk of Maternal Primary Infection and Re-Infection

Primary prevention consists of coordinated actions to prevent foreseeable problems, protect existing states of health and healthy functioning, and promote the potentials of individuals and groups in their physical and sociocultural environments over time [19]. Knowledge of CMV and its mode of transmission is the cornerstone of primary prevention, enabling women of childbearing age and in early pregnancy to apply adapted hygienic measures to avoid CMV primary infection or re-infection [20]. Despite an evolution in awareness over the last decade, the most recent studies still show significant gaps in the knowledge of pregnant women [21,22,23], which considerably reduces the effectiveness of primary prevention. At the same time, constant progress in CMV vaccines may provide a promising option to prevent maternal infections during pregnancy in the near future.

### 2.1. Primary and Non-Primary Maternal Infections

Like other DNA herpes viruses, after primary infection, CMV establishes as a latent viral infection that persists for life [24]. The two main pathways of primary maternal infection of CMV are sexual intercourse and close contact with young children [25]. Transmission can occur through direct or indirect person-to-person contact with infectious body fluids, including semen, cervical or vaginal secretions, saliva, urine, and blood products [3]. Pregnant women have been identified as a population of concern for CMV infection, and among them, those expecting their second child, as well as those who are seronegative in their first pregnancy, have been identified as being at particular risk for primary infection [26,27]. The incidence of primary infection among these pregnant women is 6%, i.e., 20 times higher than other pregnant women [28]. A meta-analysis by Hyde al. (2010) identified that daycare worker and pregnant women in contact with children shedding CMV (respectively 8.5% and 24% of annual primary infection) were at particular risk of primary infection [29]; in contrast, healthcare professionals were not particularly prone to primary infection [30]. If the infection is symptomatic, a pregnant woman will develop flu-like or mononucleosis-like symptoms that may include mild fever, rhinitis, pharyngitis, headache, fatigue, and hepatic disorders [3,31]. The transmission from child to adult is mainly through direct or indirect contact with an infected child’s saliva, urine, or tears [29,32]. Mother-to-child transmission may occur in the prenatal, perinatal, or postpartum period, but to date, only prenatal transmissions have been correlated with cCMV infection [29,33,34].

Non-primary infections may occur either due to an infection with a different strain (reinfection) or as a result of reactivation of an endogenous strain. In recent years, the epidemiological importance of non-primary infections has received increasing attention, as studies have shown that non-primary infections can cause serious congenital infections in newborns [35,36,37,38,39,40,41]. The rate of maternal–fetal transmission after re-infection is often described as low, but it may be underestimated. This may explain why a country with a high seroprevalence of CMV, such as Brazil (>70% of the population), nevertheless reported a prevalence of congenital CMV infection of about 1.1% [42]. However, the diversity of serological and molecular patterns associated with non-primary CMV infections make diagnosis difficult, requiring a better understanding of the mechanism of intrauterine transmission of CMV virus in neonates of mothers with pre-existing immunity [36].

### 2.2. Women’s Knowledge and Awareness

#### 2.2.1. Evolution of Awareness in Pregnant Women

The current literature agrees that increasing CMV knowledge and awareness among pregnant women, as well as about its mode of transmission, seems to be effective in reducing the rate of infection [43,44]. Evidence shows that current CMV prevention messages aim to reduce the probability that a pregnant woman will receive urine or saliva from young children in her eyes, nose, or mouth [45]. However, the scientific consensus regarding this level of knowledge is that it remains low overall, despite the increased warnings from specialists in recent years about the fetal consequences of such an infection. Indeed, a study conducted in Japan between 2012 and 2016 showed no change in knowledge about the virus in the population of pregnant women, who maintained an extremely low knowledge rate over the study years (7% over 6 years) [46]. A low level of CMV-related knowledge was also found among woman of childbearing age and/or pregnant woman in other studies [45,47,48,49,50]. This rate varies greatly from one country to another, with a percentage of knowledge of less than 20% reported in some studies conducted in Ireland, the USA, Holland, Japan, Saudi Arabia, and Australia [32,47,48,49,51,52,53]. Three studies conducted in Canada, the USA, and England between 2019 and 2021 reported a rate of CMV awareness among pregnant women of between 32.4% and 39% [44,54,55], while three other studies conducted in Switzerland, Italy, and France showed rates of awareness among pregnant women ranging from 39% to 60% [15,56,57]. While the rates found in the first three articles may suggest, in view of their recent publications, that women’s awareness is shifting toward a better knowledge of the virus, the last three articles stand out due to the context in which they were conducted, including an active preventive policy in the follow-up of pregnant women [15,56,57,58]. The authors who extended their previous investigation on women’s knowledge of cCMV symptoms found that women generally failed to correctly identify the symptoms associated with cCMV [49,56,59].

#### 2.2.2. CMV Awareness Compared to Other Conditions

A comparison between women’s knowledge of CMV and knowledge of other infections or diseases that can infect the fetus has been conducted by some authors. All results showed that women’s knowledge of CMV was much lower than their knowledge of toxoplasmosis or Down syndrome, which, however, have a lower incidence than CMV infection [44,48,50,53,56,57,59,60]. In the United States, the number of children born with cCMV and its long-term sequelae is estimated at 8600 per year, which is much higher than the number of children born over the same period with sequelae due to toxoplasmosis and Down syndrome (about 1000 and 4000 cases, respectively), which are diseases that are better known by women [59]. 

#### 2.2.3. Application of Hygiene Measures

Studies investigating women’s attitudes once they were informed about the risks of CMV infection during pregnancy demonstrated that participants had a positive attitude toward education, with a large majority of women requesting further information and prenatal serological testing [54,60,61]. Hygiene measures remain the most effective prevention strategy, reducing the rate of primary infection during pregnancy by four to five times [14,17,62,63]. However, women’s knowledge of key preventive hygiene measures varies from extremely low (5%) to fairly high (92%) in the literature [56,64]. Washing hands after changing a baby’s dirty nappy seems to be the most common preventive measure known and applied by the women interviewed to prevent possible CMV contamination. Rates of knowledge of hygiene measures, such as not sharing the same drinking glass, spoon, or fork with a child under five years of age, as well as not kissing on the lips and avoiding contact with nasal secretions or tears (when assessed), rank about the same in the studies analyzed, although with wide variations between studies [45,48,53,56,57,64]. It should be noted, however, that half of the studies that analyzed these different variables reported an overall knowledge rate of less than or equal to 50% [48,53,64].

Once the women had been informed of the hygiene practices to be applied, they reported an ease of application ranging from 65% to 98% regarding washing hands after changing diapers, avoiding sharing cutlery or food, and wiping runny noses or tears [45,48,49,53,54,56,57]. Ease of implementation of the preventive measure regarding not kissing infants on the lips appeared to be more difficult for the respondents to put into practice [55]. As this habit can be cultural and subject to demographic variables, particular attention should be paid to this socio-cultural aspect when teaching pregnant women to ensure that the preventive message will be optimized and delivered appropriately [32]. 

#### 2.2.4. Effectiveness of Educational Interventions

Several studies have shown that educational intervention during pregnancy decreased the frequency of activities that could expose women to body fluids, such as saliva and urine, from young children [53,62]. The advantages of visual support as an effective educational tool have been highlighted by several authors to optimize the integration of preventive messages among women [44,56,65]. In their study, Lazzaro et al. (2019) argued that 99% of women sought further information after being educated about CMV by their healthcare provider [53]. Access to CMV-specific information and knowledge of preventive measures empowers women to make the necessary changes and modifications for the safety of their babies [66,67]. The lack of information provided to women was seen by some participants as an impediment to their own free will [61].

The hypothesis that was developed several years ago suggesting a correlation between the application of hygiene measures and a low seroconversion rate [68,69] has since been supported by results of different studies [20,44,62]. In 2009, Vauloup-Fellous et al. found that careful application of hygiene measures could lead to a decrease in seroconversion rates of up to 80% (N = 5/2583 (0.19%)) [62], and more recently, Revello et al. (2015) [25] also showed that awareness and knowledge of preventive hygiene measures significantly reduced the rate of seroconversion in a sensitized group compared to a non-sensitized group (1.2% vs. 7.6%; Δ = 6.4%; 95% CI 3.2–9.6; *p*-value < 0.001). The authors concluded that primary prevention decreased the rate of congenital infection in newborns (0.9% vs. 2.5% of cumulative incidences in the sensitized and non-sensitized groups, respectively) [25]. Therefore, getting women to actively participate in the preventive process becomes crucial as early as possible in pregnancy. The first trimester has been reported to be the time when seroconversion is most likely to result in long-term fetal sequelae in case of vertical transmission [70]; the timing of awareness has indeed been considered critical by scientists. Therefore, in order to reduce the risk of seroconversion, a preventive message should be delivered in the pre- or periconceptional period [59]. Despite this, it has been shown that a large majority of pregnant woman are informed about CMV and preventive hygiene measures at their first consultation, often at the end of the first trimester [56]. This delay in awareness among pregnant women should be considered in social and health educational programs and implemented, according to some authors, from school onward in order to reach a greater number of women before the start of a pregnancy [56,59]. 

### 2.3. Healthcare Professionals’ Knowledge and Attitudes

#### 2.3.1. Role and Training of Health Professionals on CMV

The role of health professionals in primary prevention is essential as women expect to receive information on CMV, along with other pregnancy-related risks, from their health professionals [71]. However, the results of numerous studies conducted worldwide have shown that the practices of health professionals regarding CMV information and prevention differ considerably from one professional to another [21,72]. In order to understand the reasons for these divergent practices, some authors have looked at the training of future health professionals as well as the place allocated to CMV education in academic curriculum [23,73]. One study reported that gaps were identified early in the training of future physicians. Using a questionnaire to assess their general knowledge of CMV and cCMV during their first four years of training, it was shown that although there was a significant increase in medical students’ awareness of CMV during their years of study, due to a pre-clinical course on infectious disease given and the encounter with cCMV patients during clinical rotations, there was, nevertheless, a significant lack of knowledge regarding modes of transmission and available treatments at the end of the university curriculum [73]. Among the strategies investigated by different authors to improve knowledge of CMV among health professionals, early education on the virus and its mode of transmission in medical school, as well as the implementation of relevant educational materials and target education, is among the training opportunities to be developed in the future [23,73,74].

#### 2.3.2. Gaps in the Knowledge of Health Professionals Lead to Ineffective Primary Prevention

The current literature has highlighted diverse levels of knowledge gaps that interfere with the ability to deliver an effective preventive message to pregnant women. While some routes of transmission seem to be better known than others (kissing on the lips, changing diapers, and infrequent hand washing), 9% to 20% of the health professionals interviewed were not able to correctly identify the specific routes of CMV transmission [21,72,75,76]. An analysis of health professionals’ knowledge of cCMV-related symptoms showed that their knowledge rate ranged from 50% to 94%, depending on the study [21,72,77,78], and the rate of knowledge of long-term cCMV symptoms was known by 43% to 94% of the professional groups surveyed [21,72,75,77,78]. In addition, 20% to 58% of the professionals who were questioned on the subject thought that a treatment was available on the market [21,72,77,78]. A gap between knowledge of CMV and implementation of preventive message for women has been noted by several authors. About 32% to 70% of the health professionals surveyed never gave information about CMV to their pregnant patients [21,22,23,76,78,79]. The reasons given by the participants for not practicing prevention with their patients were insecurity about their lack of knowledge of risk factors and preventive measures to avoid infection [23,71,76,79], fear of being a source of anxiety for the mother [23,71,79], lack of time [79], the belief that the disease is rare [79], the fact that it is not considered a common practice [79], and the fact that it is not a mandatory recommendation [23,78]. Lack of knowledge is a recurrent theme mentioned by the participants to explain their absence of preventive discourse [71,76,79,80,81]. 

#### 2.3.3. Enhance the Knowledge of Health Professionals to Educate Pregnant Women

In order to determine whether increased knowledge of the virus by health professionals could have a positive impact on preventive messages delivered to pregnant women, studies were conducted on the same group of professionals before and after attending an educational program on the virus [23,71,80]. One of the studies found that the rate of correct responses related to CMV increased by 26.9% (*p*-value < 0001) and that 100% of the participants intended to increase their antenatal counselling on CMV after attending the program [80]. Other authors also reported this trend [23,71]. A study carried out in France with a follow-up that was six years apart and with no specific educational program offered to the participants, nevertheless, reported an improvement in the knowledge of health professionals concerning the route of transmission of the virus, its maternal and neonatal symptoms, and its long-term neonatal sequelae [21], as well as an increase in the practice of recommendations concerning preventive hygiene measures. The researchers noted an improvement of more than 36% in knowledge and a significant association between the level of knowledge and the delivery of preventive messages (*p*-value = 0.005; OR = 2; CI (1.2–3.1)) [21]. The improved knowledge of health professionals was correlated with a better knowledge of the virus, the importance of primary prevention to decrease the rate of seroconversion during pregnancy, and the update of national guidelines giving a clear direction to the practice [21,23,80]. 

### 2.4. Vaccines

The development of an effective vaccine is highly prized by the scientific community. However, despite a clear need, progress toward its realization has been slow [82]. Frequent reinfections and reactivations of the virus are the main obstacles to the effectiveness of vaccines. These indicate not only that the immunity naturally induced by the virus is not perfect to protect against re-infection [83], but also that the immune correlates of protection (CoPs) are still unclear, leading to major difficulties for researchers to obtain potent neutralizing antibodies (NAbs) [84]. Therefore, in order to maximize the effectiveness of vaccine protection, researchers should consider exploiting both the precepts of innate and adaptive immunity [85]. Indeed, a global understanding and analysis of the mechanisms related to the immune responses developed during CMV infection, such as its replication mode or the ability of the virus to evade certain immune responses, would allow the identification of protective immunity components, thereby opening the possibility to mimic these immune responses through induced immunization [85].

Innate immune responses include the natural killer (NK) cell response, which is the first line of defense against viral infection by allowing rapid antiviral functions to be performed by the host [84,85,86]. Currently, in the context of CMV infections, these cells are the best documented innate memory response on the subject [84]. Conventional vaccination uses humoral and cellular responses mediated by B and T lymphocytes to provide protection against the virus [83,85]. In addition, responses developed during natural infection can block infection of different cell types, making NAb induction a major target for vaccine development [87,88]. In a large cohort study of 3461 women in the United States, the rate of newborns with cCMV was 3% in those born to initially seronegative mothers, compared to 1% in those born to mothers immunized prior to pregnancy. The authors of this study concluded that pre-existing maternal immunity could be associated with a reduced risk of vertical transmission of CMV to the fetus [89]. Recent studies also demonstrated that specific T-cell responses could be correlated with the protection from vertical transmission in congenital CMV infection [90,91].

Various strategies for developing an effective vaccine have been investigated by the scientific community in recent years. Vaccines that are currently under development are mainly based on the following designs: live-attenuated and disabled-infectious single-cycle, adjuvanted recombinant protein, DNA, mRNA, virus-like particle, viral vectored, and peptide vaccines.

#### 2.4.1. Live-Attenuated and Disabled-Infectious Single-Cycle Vaccine

Currently, five of these vaccines are being tested in phase I and II trials [85]. It has been established that this type of vaccine induces both humoral and cellular responses [83,84,85]. Although no safety issues have emerged from these early phase trials, this vaccine design, which establishes latency and/or undergoes productive replication in the immunized host, carries an unacceptable level of risk that is of concern to the scientific community [92]. Therefore, replication-deficient CMV vaccines have been developed in parallel to increase the safety of live-attenuated vaccines [85]. The promising results obtained led to the inclusion of a number of these vaccines in a phase II clinical trial aiming at studying the efficacy of the two- or three-dose vaccine in CMV-negative adolescents and women of childbearing age. Although well tolerated and immunogenic, the vaccine efficacy was only 42.4% in the three-dose group and 32.0% in the two-dose group, which may be considered too low to proceed to phase III trials [85].

#### 2.4.2. Adjuvanted Recombinant Protein Vaccine

Currently, six vaccines with this design are in phase I and II trials [85]. The CMV glycoprotein B (gB) has been identified as a major vaccine target due to its essential role in the fusion of virions with target cells and its ability to elicit NAb and non-NAb responses [85,93]. In phase I, the most promising of these vaccines was conducted in children aged 12 to 35 months. The results obtained on this population, compared to those obtained on an adult population, showed that not only was the vaccine well tolerated in toddlers, but also that their gB-specific antibody response was significantly higher when compared to adults [85]. This suggested that CMV vaccination in the early years of life may be a feasible solution to reduce CMV infection [85]. Subsequently, the vaccine was tested in phase II clinical trials as a strategy to prevent congenital infection. The vaccine demonstrated an efficacy rate ranging from 45% [94] to 50% [95] against CMV acquisition, but it did not achieve a significant reduction in the risk of infection in one of these trials and did not proceed to the next stage due to this moderate efficacy [85].

#### 2.4.3. DNA Vaccine

Two DNA vaccines are in phase I, II, and III testing. This type of vaccine is composed of plasmids coding for vaccine antigens. These DNA vaccines use the genes coding for the tegument protein pp65 and/or the envelop gB [85]. In phase I clinical trials, one vaccine not only demonstrated good tolerance, but also elicited antigen-specific T cell responses and NAbs in seronegative subjects [84,85,96]. In phase II trials, CMV viremia in hematopoietic stem cell transplant (HSCT) recipients was significantly decreased in plasma after vaccine administration, without demonstrating significant differences in other parameters, such as T-cell production, between the vaccine and placebo groups [84,96]. One vaccine is currently being studied in HSCT donors and recipients in phase II and III clinical trials (Astellas), but the results to date have not demonstrated significant improvement in overall survival and reduction in terminal CMV disease [85].

#### 2.4.4. Messenger RNA Vaccine

A messenger RNA (mRNA) vaccine is an antigen vaccine in the form of a modified mRNA encapsulated in lipid nanoparticles (LNPs) or liposomes. These mRNA vaccines use the genes coding for the tegument protein pp65, gB, and/or the pentamer complex (PC). It leads the immune system to produce responses against its corresponding antigen [82]. Currently, two of these vaccines are in phase I, II, or III testing [85]. ModernaTX, Inc. (Cambridge, MA, USA) tested two mRNA vaccines: the mRNA-1647 vaccine (composed of one mRNA from gB and five mRNAs from PC) and the mRNA-1443 composed of pp65. These vaccines were evaluated in healthy adults in phase I clinical trials. The mRNA-1647 showed a persistent immune response six months following a third dose and a high NAb titer specifically on epithelial cells [85,96]. NAb titers of CMV-seronegative vaccinees increased 2.8- to 17.0-fold in epithelial cells and 0.8- to 5.0-fold in fibroblasts when compared to CMV-seropositive individuals. In addition, the NAb titer of CMV seropositive vaccinees increased 4.0- to 7.1-fold in fibroblasts and 13.4- to 40.8-fold in epithelial cells [85]. After some modifications concerning mainly the ratio of mRNA components, the optimization of the manufacturing process, and the switch from liquid to freeze-dried potency, one of the vaccines was able to enter phase II clinical trial. Since phase II trials demonstrated good tolerability and antigen-specific functional responses, thus supporting the potential of this vaccine candidate in preventing CMV infection, healthy adult women aged 18 to 40 years are currently being enrolled in a phase III clinical trial (Moderna, Cambridge, MA, USA) [97].

#### 2.4.5. Virus-like Particle Vaccine

A single virus-like particle (VLP) vaccine is currently being tested in phase I [85]. Viral-like enveloped particles (eVLPs) are protein structures that simulate wild-type viruses [98]. However, they lack a viral genome, making them non-infectious, and, thus, may represent safer vaccine candidates [93,96]. The response of this vaccination would be the production of neutralizing antibodies against envelop glycoproteins, such as gB, gH, gL, and PC. Indeed, these glycoproteins are involved in the attachment/entry of the virus in human cells, and neutralizing antibodies could prevent the infection and spread to placental cells [99]. In 2018, both the aluminum phosphate-adjuvanted (APA) and unadjuvanted vaccines were tested in CMV-negative adults in a phase I clinical trial. When compared to the placebo group, the vaccines were well tolerated across different vaccine doses. The 2 μg dose group with APA showed the highest gB and NAb-specific antibody responses. In 2019, the investigators announced plans to test higher doses of the vaccine in a phase II clinical trial [85].

#### 2.4.6. Viral Vectored Vaccine

Viral vector vaccines are recombinant attenuated viral vectors encoding vaccine antigens. Five of them are currently being tested in phase I and II clinical trials [85]. This vaccine approach uses a heterologous viral vector to deliver CMV-encoded immunogens. Because viral vectors are unable to fully replicate when administered to humans, they are highly attenuated yet effectively deliver one or more viral antigens [92]. Their main shortcoming is the presence of pre-existing vector immunity, which potential development could be a problem for this type of vaccine [82]. In preclinical studies, viral vectored vaccines using envelop glycoproteins are promising targets. Both gB and PC vaccine candidates lead to neutralizing antibody response, and a good strategy could be the association of several glycoproteins [100]. The outcome measures generally target T-cell responses, and CMV-specific T-cell antigens are usually included in a viral vector vaccine [84]. Recently, in phase I clinical trials, one of these vaccines (CMV-MVA triplex vaccine of the City of Hope Medical Center) have not only demonstrated a safe profile among HSCT participants treated with the vaccine, but also the ability to elicit robust cytotoxic T lymphocyte (CTL) responses through donor vaccination in CMV-positive recipients [85,101]. Indeed, CMV-specific CD137^+^CD8^+^ T cells were significantly higher (*p*-value < 0.0001 and *p*-value = 0.0174, respectively) in recipients who received an HCT from a Triplex-vaccinated matched related donor than an unvaccinated donor (control cohort) [101]. The promising results of the cellular and humoral responses obtained in phase I have enabled some of these vaccines to enter phase II clinical trials [85,92].

#### 2.4.7. Peptide Vaccine

Research on peptide-based vaccines has focused on protection against CMV disease in HSCT recipients, rather than on prevention of congenital CMV infection [96]. Only one of these vaccines has been tested in clinical trials [85,102]. One of main advantages of this peptide vaccine, which includes viral peptide antigens, is the ability to stimulate an epitope-specific T cell response [84]. The phase I trial in seronegative patients with end-stage renal disease, who were awaiting renal transplantation, demonstrated the safety and immunogenicity profile of the vaccine. Five out of ten patients (50%) developed an immune response, and 40% of patients developed CMV-specific T cell responses induced by these prophylactic vaccinations [102]. The function and kinetics of the vaccine’s CMV-specific T cell immunity were recently studied in a phase II clinical trial [85].

#### 2.4.8. Correlates of Protection

One of the main strategies advocated by the scientific community to optimize CMV vaccine development is to identify immune CoPs [103]. These immune markers are associated with a reduction in the incidence of infection or clinical disease [92,104,105], and they could be used to identify potential vaccine targets, as well as to guide vaccine development and refinement, in order to predict vaccine efficacy in different settings and to inform vaccination policy and regulatory decisions [92]. Identification of these CoPs is of particular importance for CMV vaccine design as natural CMV infection grants only partial protection against reinfection and vertical transmission [106]. Future vaccine designs should consider the characteristics of the virus, such as optimal antigenic targets, while including an induction of cross-protection between different CMV strains, and appropriate correlates of protection [107]. At present, defining preclinical efficacy by a specific immune response triggered by a vaccine is probably not sufficient to predict efficacy in human trials and may explain the moderate efficacy of vaccines tested in clinical trials to date [82]. Evidence suggests that multiple antigenic targets will be needed and that NAbs alone are not sufficient for protection [85]. Therefore, to be effective, CMV vaccines will not only need to elicit immune responses to be even more protective or different from those elicited by natural infection, but the development of protective CMV vaccines will also likely require rational vaccine design, which can be guided by immune CoPs [105]. Until a complete understanding of the evasive mechanisms of the CMV cellular response is achieved, it will remain essential to strengthen the role of awareness in the primary prevention of CMV infections [108].

## 3. Secondary Prevention: To Reduce the Risk of Maternal–Fetal Transmission

Secondary prevention aims to reduce the impact of a disease that has already occurred. It consists of detecting and treating the disease as early as possible to stop or slow down its progression and prevent long-term problems [109]. Secondary prevention aims to avoid or reduce the risk of maternal–fetal transmission in case of proven CMV seroconversion in the mother [110]. The literature on secondary prevention of cCMV focuses on two complementary topics: maternal serological screening [111,112] and prophylactic therapies, including administration of hyperimmune globulin [24,113,114] or antiviral therapy [115]. Currently, practices regarding the application of these methods are site-specific as the effectiveness of these methods is still considered controversial [116]. This discrepancy in practice is concerning as the screening strategy will also influence the prognosis work-up and the possibilities of prophylactic treatment [117].

### 3.1. Diagnosis of Maternal Infection and Screening Strategies

Clinical diagnosis of maternal CMV infection is unreliable as it is only symptomatic in 8 to 10% cases. Even if they are present, the symptomatic, clinical signs are usually nonspecific and various [31]. Therefore, diagnosis of CMV primary infection during pregnancy mainly relies on serology, either based on seroconversion (negative IgG test that becomes positive) or the more frequently detection of specific CMV-IgG and IgM, which are associated with CMV-IgG avidity in case of positive CMV-IgM [118]. Even if the sensitivity and specificity of specific CMV IgM are good (>90%), they are far less reliable for the diagnosis of primary infection [118] (positive predictive value of 15–40%, depending on if it is universal or target screening) [119]. Indeed, CMV-IgM can possibly indicate an acute or a recent infection, but most often, positivity is due to other causes, such as long-term persisting IgM, cross-reaction, secondary CMV infection, or nonspecific stimulation of the immune system [112]. This lack of specificity in predicting recent primary infection can lead to misinterpretation of results, causing an assumption of a primary infection. Consequently, diagnosis of primary infection cannot rely only on a positive IgM test result. Furthermore, although a low CMV IgG avidity confirms a recent primary infection while a high CMV IgG avidity excludes it, there is a large gray area in interpreting the results where an intermediate IgG avidity cannot fully exclude a recent primary infection [119]. Reported clinical performances for CMV IgG avidity, specificity, and sensitivity range between 90 and 100%, depending on the assay [120,121]. CMV-IgG levels are initially of low avidity but will mature to high avidity at 2–4 months after a primary infection [122,123]. Therefore, a low avidity result associated with a positive IgM antibody indicates a primary infection that has occurred within the previous three months, allowing a more accurate diagnosis of the timing of the primary infection during or shortly before pregnancy (Figure 1) [124].

Recent improvement in screening techniques has greatly increased the quality of serological testing and has been advanced as an argument for universal screening early in pregnancy and in women willing to conceive [117]. Serological screening is minimally invasive and well received by the pregnant population (only 3% would refuse to be diagnosed). However, serological screening is only reliable for detecting primary infections. In most patients with a confirmed CMV non-primary infection during pregnancy (positive IgG before pregnancy with re-activation of viremia during pregnancy and/or cCMV infection in the newborn), serology fails in detecting CMV re-infection or reactivation [35,36,126,127]. Indeed, >50% women immunized before pregnancy and delivering an infected baby have stable CMV-IgG titers, negative CMV-IgM, and negative PCR in serum. Moreover, increased CMV-IgG titration, as well as a high CMV-IgG avidity index, and/or a positive CMV-IgM can be attributed to other clinical situations that are more frequently encountered than CMV non-primary infection. Detection of CMV-DNA by PCR in whole blood can indicate both primary and non-primary CMV infections [128]. Viremia in non-primary infection during pregnancy, as in immunocompetent non-pregnant individuals, seems to be transient, and the viral load can be very low, thus limiting its detection [129]. However, when CMV-DNA is detected by PCR in a pregnant woman with positive IgG prior to pregnancy, this could indicate a non-primary infection. Differentiating reactivation from reinfection is impossible using standard PCR or serologies. However, the appearance of a new antibody with specificity against polymorphic epitopes of CMV, as detected by strain-specific ELISA, may indicate reinfection [40,130]. The diagnosis of non-primary infection during pregnancy remains challenging and explains why routine molecular serologic analysis is not considered in this context [36].

As previously established, the presence of CMV viruria can be detected by PCR after a primary infection or reactivation/reinfection. CMV viruria is then intermittent and variable, and it can be present from a few weeks to a few months after infection [131,132]. After a primary infection during pregnancy, some studies suggest that a positive viruria is associated with a slightly higher rate of maternal–fetal transmission, but in case of cCMV infection, the risk of fetal sequelae does not appear to be increased [133,134]. However, very few studies have evaluated the use of CMV PCR in urine for the detection of CMV infection in pregnancy, and its input is, therefore, not known [135]. In particular, its use as a screening strategy during the first trimester of pregnancy for non-primary infection has not been evaluated.

Although various reasons have been advanced for questioning the relevance of routine serological screening of pregnant women (difficulty of interpretation, lack of treatment available, etc.) [4], some specialists advocate for informing all pregnant women about the possibility of serological screening in early pregnancy, in order to optimize the detection and follow-up of congenital infections [117,136]. If this screening is desired by a pregnant woman and/or locally recommended, it should be offered as early as possible during pregnancy, and ideally pre-conceptionally to simplify the interpretation of results [117,136]. Nevertheless, it is important to note that the discussion on serological screening in early pregnancy must consider the availability of screening, prenatal diagnosis, treatment, and termination of pregnancy at reasonable costs in the countries concerned, and that the overall value of such screening also depends on the local epidemiology of CMV. Thus, it seems difficult to propose universal recommendations for CMV screening in pregnancy.

A few cost-effectiveness studies have investigated serological screening for prenatal detection of congenital cytomegalovirus following maternal primary infection. The first studies, which did not include recent data on the efficacy of valaciclovir in preventing maternal–fetal transmission, showed that routine CMV serology screening could be cost-effective if antenatal treatment was significantly effective in reducing the risk of neonatal disease [137,138]. Since then, a meta-analysis of the results of three recent studies has concluded that valaciclovir is effective for secondary prevention of maternal–fetal transmission of CMV [115,139,140,141]. An American cost-effectiveness analysis, based on a cost of USD 100,000 per quality-adjusted life year (QALY), demonstrated that universal first-trimester serological screening for primary maternal CMV infection is not cost-effective as it resulted in only 14 fewer children being affected with cytomegalovirus per 100,000 pregnancies when compared to usual care [142]. A French study assessed the cost-effectiveness of prenatal detection of congenital cytomegalovirus following maternal primary infection during the first trimester within standard pregnancy follow-up or involving population-based screening (serological testing at 7 and 12 weeks of gestation). CMV serological screening followed by valaciclovir prevention might prevent 58% to 71% of severe congenital CMV cases for a cost of EUR 38 per pregnancy [112]. In this study, the cost of postnatal sequelae and care for these children was not considered. More recently, a Japanese study estimated that systematic screening for first-trimester primary infection associated with valaciclovir would represent an additional cost of USD 6604/QALY when compared to the absence of screening for CMV [143]. These new data could fuel health authorities to develop clinical guidelines on the identification of congenital CMV infection according to public health policies. The different screening strategies can only identify maternal primary infections, and non-primary infections will not be detected. It is, therefore, important to specify that these cost-effectiveness analyses are only valid in similar populations, for which the seroprevalence rates are similar (i.e., approximately 50% of women of childbearing age [5,144]). Further cost-effectiveness studies are needed to specify the most appropriate strategy, but it would seem appropriate that such serological screening, if it takes place, should focus on the period of greatest risk (i.e., first trimester) and be repeated so that valaciclovir treatment can be initiated as soon as possible after seroconversion.

### 3.2. To Reduce the Risk of Maternal–Fetal Transmission in Case of Maternal Infection

Maternal–fetal transmission of CMV is the result of transplacental passage of the virus, which then replicates in multiple embryonic or fetal tissues [3]. Vertical transmission from the mother to the fetus in primary infection occurs in approximately 10% to 70% of cases [3,24,31,145]. A recent meta-analysis of 17 studies that analyzed vertical transmission rates in relation to gestational age found that the risk of vertical transmission is strongly correlated with the gestational age at which the primary infection occurs (Table 1) [146]. Indeed, the risk of vertical transmission of CMV in utero increases with advancing gestational age, but the risk of fetal/neonatal sequelae is inversely related to the gestational age at the time of infection [70,146]. Although the rate of transmission is high in the third trimester, no study to date has been able to show severe cCMV symptoms in newborns infected at this gestational age [4].

The rate of non-primary maternal–fetal transmission is not known as the diagnosis of non-primary infection in pregnant women is not reliable and usually not performed. In the case of proven maternal primary infection, cCMV infection can be diagnosed by amniocentesis at 17 weeks of gestation or possibly by chorionic villus sampling performed at 11–14 weeks of gestation. In order to achieve optimal sensitivity of these diagnostic tests (>95%), a consensus time frame of six to eight weeks after a primary infection should be respected (unless there are ultrasound signs) [147]. The sensitivity of CMV detection in amniotic fluid and in trophoblast samples obtained by amniocentesis or chorionic villus sampling is, respectively, 45–80% [148] and 50% [149]. The specificity approaches 100% for both.

A recent double-blind, randomized controlled trial found that oral valaciclovir is effective in reducing the rate of fetal cytomegalovirus infection after an early maternal primary infection acquired either periconceptionally or during the first trimester of pregnancy. In this randomized controlled trial, treatment was initiated after periconceptional (i.e., within four weeks before the last reported menstrual period and up to three weeks of gestation) or first-trimester primary infection, and the primary endpoint was the presence or absence of CMV detected by amniocentesis performed at around 21 weeks of gestation [115]. Forty-five patients per arm were included and received valaciclovir at a dose of 4 g × 2 per day, and the results showed a 70% reduction in the risk of maternal–fetal transmission (11% in the treated group vs. 30% in the placebo group, OR = 0.29; 95% CI 0.09–0.90). In the subgroup analysis, this difference was still significant among the patients who seroconverted in the first trimester (11% vs. 48%; *p*-value 0.020) but not in the group of patients who seroconverted periconceptionally (12% vs. 13%, *p*-value = 0.91). A lower transmission rate and a higher delay from seroconversion to initiation of treatment in the periconceptional group could be the main factors limiting the effectiveness of valaciclovir. Additionally, two retrospective case–control study found similar results [139,141] (Table 2). Faure-Bardon and al. (2021) also found a decrease in maternal–fetal transmission when they assessed CMV PCR results at the time of amniocentesis. This study involved a cohort in a single centre offering CMV serological screening at 11–14 gestational weeks [139]. Egloff and al. (2022) evaluated the efficacy of valaciclovir in all trimesters of pregnancy and also found an efficacy of valaciclovir after using a propensity score according to the trimester of seroconversion [141]. Valaciclovir was significantly associated with an overall reduction in the rate of transmission (odds ratio, 0.40 (95% CI, 0.18–0.90); *p*-value = 0.029), with a similar trend also observed in patients who seroconverted in the second trimester of pregnancy (statistically not significant). However, the use of valaciclovir beyond the first trimester remains debated in view of the low risk associated with congenital infections in the second and third trimesters of pregnancy. Furthermore, Egloff et al. (2022) showed that valaciclovir was more effective in patients with positive maternal viremia (assessed by PCR using maternal blood) [141]. The data suggest that maternal–fetal transmission is likely to occur in the acute phase after seroconversion when viremia is most prominent, which is variable and can last from several weeks to several months [150]. Therefore, the time from seroconversion to treatment initiation is a major prognostic factor for the effectiveness of valaciclovir in preventing maternal–fetal transmission [115]. It should be noted that the dosage of valaciclovir in these three studies is four times higher than that used as its typical dosage (a dosage of 8 g/d comes from studies showing its effectiveness on CMV reactivation and reinfection in renal transplant patients [151]). Side effects for pregnant women could be more frequent with this dosage. For example, among these three trials, two cases of acute renal failure (approximately 1% of patients) were identified after the initiation of valaciclovir (8 g per day) and resolved spontaneously after the discontinuation of treatment [139]. These data highlight the importance of clinical and laboratory monitoring throughout the duration of treatment. For the moment, none of the data from these three studies are alarming for exposed fetuses, but further evaluation through long-term pharmacoepidemiological studies is crucial to ensure the safety of valaciclovir at this dosage.

Other antiviral treatments could be discussed. Ganciclovir has not been used due to concerns about teratogenicity and toxicity to fetal germ cells [152,153], although it has been shown to be particularly effective on CMV [154]. Letermovir, which has specific anti-CMV activity and is well tolerated, is currently under evaluation to treat infected fetuses [153].

Another alternative is the administration of CMV hyperimmune globulins to infected pregnant women during their first trimester of pregnancy. Although some recent randomized controlled trials investigating the efficacy of hyperimmune globulins did not find a significant reduction in the incidence of cCMV infection [116,155,156], an observational study with the administration of hyperimmune globulins twice a month showed promising results in decreasing mother-to-child transmission rate [157]. However, the fact remains that hyperimmune globulin therapy during pregnancy is much more expensive than valaciclovir, and it may cause severe allergic reactions in rare cases (1/203 in the study by Hughes et al. [111]).

The recent emergence of scientific evidence, which attributes a number of benefits to secondary prevention through the administration of valaciclovir to reduce the risk of vertical transmission by up to 70% [117], has not only brought a therapeutic management perspective to primarily infected women, but it has also led some researchers to recommend routine early detection of CMV [24,115,117,139]. The existing controversy between primary prevention through hygiene promotion, which is advocated by some authors [20], and secondary prevention through systematic serological screening [117] must be determined by patient follow-up policies that are implemented within health care systems and by the availability of therapeutic offers. Treatment strategies must, therefore, be established according to the practice settings.

## 4. Tertiary Prevention: To Reduce the Risk of Symptomatic cCMV Infections

Finally, tertiary prevention aims to lower the impact of a disease that has long-term effects. It involves developing follow-up or treatment to improve as much as possible the ability to function, quality of life, and life expectancy [109]. Tertiary prevention for cCMV is mainly based on surveillance for signs of fetal infection caused by the virus [7]. In the case of prenatal diagnosis of cCMV infection, recent advances in fetal medicine have made it possible to propose a prognosis work-up based on ultrasound, MRI [158,159,160], and fetal blood sampling [161], and a curative in utero treatment has been developed aiming to reduce the risk of sequelae in fetuses exhibiting mild to moderate signs of cCMV infection.

### 4.1. Prenatal Signs of Congenital CMV Infection and Prognosis

Determining the neonatal prognosis once fetal infection has been confirmed is crucial to establishing the best possible management, but giving an accurate prognosis is complicated [4]. The prognosis depends on three main variables: the time of infection, the presence and type of fetal abnormalities, and laboratory parameters [4]. There are various methods available to enable the most accurate prognosis to be made [158,162,163]. Since the diagnosis of cCMV is often an incidental finding during routine examinations, the first signs described are often those observed on prenatal ultrasound. Cerebral abnormalities appear to be the main ultrasound prognostic indicator [164]. Magnetic resonance imaging (MRI) is a valuable tool for high-risk cases as it allows a more accurate investigation of the abnormalities detected by ultrasound [160,165,166].

Ultrasound signs may take several weeks or months to appear following the natural history and progression of the disease. It is, therefore, necessary to repeat ultrasound during pregnancy after a CMV infection (every 2–4 weeks). The purpose of ultrasound is to determine fetal prognosis. The negative predictive value of ultrasound has been estimated at 90% [7]. In a Belgian study, the sensitivity was only 37%, but the estimates were based partly on autopsies of medical terminations of pregnancy, which do not necessarily correlate with neonatal symptoms. In addition, there was no information on the term of pregnancy when an ultrasound was performed (CMV-related diseases are progressive during pregnancy), which might modify the sensitivity and negative predictive values, thus leading to an underestimation [167]. Altogether, it seems that ultrasound has a good negative predictive value but a lower sensitivity and positive predictive value. Prenatal ultrasound features can be labeled as extracerebral and cerebral findings. Cerebral abnormalities are the main prognostic factor [7,168]. In the presence of normal prenatal ultrasound and MRI, the outcome of a primary congenital CMV infection is generally favorable [160]. However, this is not indicative of hearing outcomes. The extracerebral findings are placentitis (thick and heterogeneous placenta), oligo- or polyhydramnios, hepatosplenomegaly, hyperechogenic bowel, miliary thin calcification, and SGA < 10th centile. These are non-specific features. Furthermore, CMV may be responsible for fetal anemia. A dilated heart could reflect mild anemia, fetal cardiomyopathy, or, more rarely, part of a full-blown hydrops fetalis. Extra cerebral findings do not allow the prediction of sensorineural hearing loss (SNHL) or neurodevelopmental abnormalities and can be seen following a maternal primary or non-primary infection at any trimester of the pregnancy, which can be isolated or associated with brain damage.

Brain lesions develop more often following a maternal primary infection in the first trimester of pregnancy. The severity of the cerebral ultrasound features can be graded from mild and good prognosis to severe or very severe (Table 3) [159,167,168].

MRI is not useful in cases of major cCMV fetopathy observed on ultrasound and for which a poor prognosis is known, but it can help to clarify the prognosis of infected fetuses without severe signs at ultrasound [171]. Therefore, MRI is a fetal imaging technique complementary to ultrasound, which is mainly contributive in the third trimester to examine the cortex, including its gyration and the white matter [172,173]. However, it should be used with caution because some findings do not have clinical implications and prognosis is unknown (White matter signal abnormalitites, for example). According to Benoist and al. (2008), the combination of ultrasound and MRI would allow the detection of brain lesions with a sensitivity of 95% [159]. Fetal cerebral MRI appears to be indicated in the case of fetuses infected during the first trimester of pregnancy that do not show severe damage on ultrasound.

In cases of proven fetal infection without associated ultrasound abnormalities, further analysis of the viral load in the amniotic fluid may help to establish a prognosis by distinguishing those who will be asymptomatic at birth from those who will be at greater risk of developing sequelae [7,174]. Many authors have reported higher median viral DNA values in the amniotic fluid of symptomatic fetuses than in asymptomatic fetuses using this method. Despite this, some authors have reported a high risk of overlapping results between the two groups [147,175,176]. Indeed, the results demonstrated that while a low viral load in the amniotic fluid was frequently associated with asymptomatic fetuses, a high viral load could be associated with both symptomatic and asymptomatic fetuses [161]. The reasons given for the lack of reliable association between viral load and fetal prognosis can be attributed to secondary variables, such as gestational age, timing of maternal primary infection or timing of vertical transmission, and timing of amniocentesis [162]. In order to refine this fetal prognosis, after a positive amniocentesis for CMV infection, some authors have focused on fetal blood analysis as a complementary input [177]. An analysis of selected hematological and biochemical markers in symptomatic and asymptomatic fetuses provided satisfactory predictive indications, suggesting that the combination of specific and aspecific marker analysis, together with the results found through the previously mentioned screening methods, could be a major asset in identifying fetuses at risk of cCMV disease and fetuses with a favorable prognosis. This could also be an added value when assisting parents in making decision about the outcome of the pregnancy during the antenatal period [161]. Indeed, some studies have shown that thrombocytopenia (less than 50,000/mm^3^ or 100,000/mm^3^) [161,165,178], high viral load (greater than 4.5 log10 copies/mL) [7,179], high B-2 microglobulin (greater than 11.5 mg/l) [161,178], and elevated hepatic markers (GGT ≥ 151 IU/L) [178] are associated with an increased risk of symptomatic neonates. A retrospective study of 82 fetuses infected after the first trimester of pregnancy highlighted the importance of a combined analysis of different predictors to achieve an optimal predictive value for fetal prognosis. Indeed, the researchers found that at the time of prenatal diagnosis, the negative predictive value of ultrasound for symptoms at birth or termination of pregnancy was 93%. Meanwhile, by pooling the variables, they were able to show that the combined negative predictive values of ultrasound and viral load in amniotic fluid and the combined value of ultrasound and fetal blood parameters were 95% and 100%, respectively. The association of either a high fetal blood viral load or a low platelet counts with non-severe ultrasound features had a positive predictive value of 79% compared to 60% for non-severe ultrasound features alone. The data on beta-2 microglobulin and hepatic markers are limited with lower evidence and have less interest shown for their evaluation as part of the fetal prognosis compared to ultrasound alone [7].

### 4.2. Preventing the Risk of Sequelae in Fetuses with Mild to Moderate cCMV Infection

Fetuses with positive PCR on amniotic fluid and mild to moderate signs on mid-trimester or third-trimester ultrasound can be treated with transplacental valaciclovir to reduce viral replication in the fetoplacental compartment [165,180,181], which could lead to improved perinatal and long-term outcomes, according to a phase II observational study [181]. The recommended treatment for infected and symptomatic neonates is valganciclovir, which rapidly achieves negative viral loads and improve hearing and developmental outcomes [182]. Despite its ability to cross the placental barrier and its use in the second and third trimesters of pregnancy reported in few cases, its use in pregnancy remains limited due to concerns regarding teratogenicity, bone marrow toxicity, and increased mutagenicity in exposed patients [154,183,184,185,186,187].

Prophylactic administration of hyperimmune globulin does not appear to be effective in reducing maternal–fetal transmission but may have a favorable effect on symptomatic forms of cCMV infection [188]. Improving neonatal prognosis by reducing the number of newborns who are born symptomatic with cCMV is the main goal of perinatal management of proven fetal infection; however, monitoring the evolution of children born asymptomatic remains critical as 10–15% of them will develop a CMV-related health problem in their first years of life [189].

## 5. Conclusions

While the various candidates for a future CMV vaccine are still being tested, education of pregnant women in early pregnancy remains the most effective primary prevention. Awareness of preventive hygiene measures should be standardized and carried out in the preconception period and as early as possible during pregnancy to avoid maternal primary infection or reinfection in early pregnancy. Depending on the health systems, local CMV epidemiology, and their risk factors, pregnant women should be informed about the possibility of screening for primary CMV infection in early pregnancy and being offered treatment with valaciclovir to reduce the risk of maternal–fetal transmission if infection occurs, especially if this approach proves to be cost-effective in future studies. Finally, estimating the prognosis of infected fetuses (based on gestational age at exposure and imaging and fetal blood sampling results) and proposing prenatal treatment with valaciclovir to increase the chance of being asymptomatic at birth for those with mild to moderate infection could also be effective in reducing the burden of cCMV infections. The safety and efficacy of currently available treatments, such as valaciclovir and hyperimmune globulin, as well as other promising antiviral drugs (letermovir), should continue to be evaluated in future studies.

## Figures and Tables

**Figure 1 viruses-15-00819-f001:**
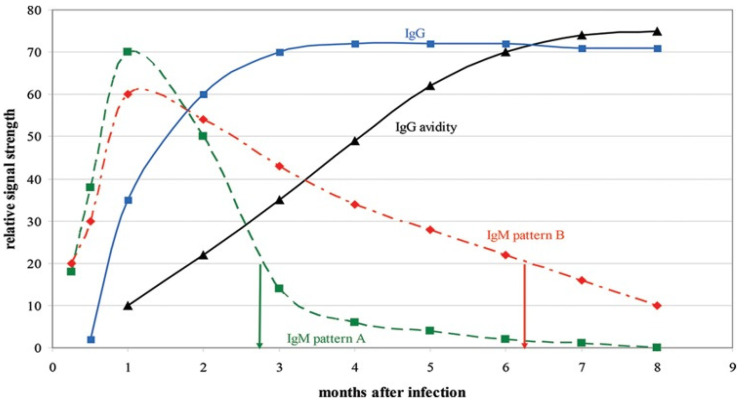
Evolution of CMV IgM, IgG, and IgG avidity levels over time following primary cytomegalovirus infection (Prince et al. 2014) [125]. Reprinted with permission from Prince et al. (2014). Copyright 2014, American Society for Microbiology. Note. CMV IgM, IgG, and IgG avidity levels show relative changes over time following primary CMV infection. IgM pattern A represents the typical IgM response pattern, whereas IgM pattern B represents long-term IgM persistence. In a CMV IgG-positive individual, an IgM-positive result of 20 indicates an infection around 3 months ago if the individual exhibits IgM pattern A, but it indicates around 6 months ago if the individual exhibits IgM pattern B. By employing CMV IgG avidity testing, the correct time since infection can be determined: a low-avidity result (expected to be about 30 based on this figure) indicates a primary infection about 3 months ago, whereas a high-avidity result (expected to be about 70) indicates a primary infection more than 6 months ago.

**Table 1 viruses-15-00819-t001:** Pooled rate of vertical transmission and fetal sequelae according to gestational age at the onset of primary infection (from Chatzakis et al. 2020) [146].

	Transmission Rate	Sequelae If Fetus is Infected ^a^
Preconception ^b^	5.5% (95% CI: 0.1% to 10.8%)	N/A
Periconception ^c^	21% (95% CI: 8.4% to 33.6%)	28.8% (95% CI: 2.4% to 55.1%)
First trimester	36.8% (95% CI: 31.9% to 41.6%)	19.3% (95% CI: 12.2% to 26.4%)
Second trimester	40.3% (95% CI: 35.5% to 45.1%)	0.9% (95% CI: 0% to 2.4%)
Third trimester	66.2% (95% CI: 58.2% to 74.1%)	0.4% (95% CI: 0% to 1.5%)

Adapted with permission from Chatzakis et al. (2020). Copyright 2020, Elsevier Inc. Note: ^a^ Fetal sequelae are defined either as the presence of neurological symptoms at birth or termination of pregnancy due to CMV-associated sonographic or MRI findings from the central nervous system. ^b^ Preconceptional period is broadly defined as three months before the last menstrual period. ^c^ Periconceptional period is broadly defined as the period between the four weeks before and six weeks after the last menstrual period.

**Table 2 viruses-15-00819-t002:** Vertical transmission rate according to gestational age at the onset of primary infection in different studies evaluating valaciclovir for the secondary prevention of maternal–fetal transmission of CMV [115,139,141].

	Valaciclovir			No Treatment		
Authors	Shahar-Nissan and al., 2020 [115] *	Egloff and al., 2022 [141]	Faure-Bardon and al., 2021 [139] *	Shahar-Nissan and al., 2020 [115] *	Egloff and al., 2022 [141]	Faure-Bardon and al., 2021 [139] *
Number of patients	45	59	65	45	84	204
Total	11%	19%	12%	30%	40%	29%
Periconceptional	11%	7%	4%	13%	10%	8%
1st trimester	12%	22%	19%	48%	41%	44%
2nd trimester	n/a	25%	n/a	n/a	52%	n/a

Note. * In these studies, maternal–fetal transmission rate of CMV was assessed by CMV PCR results on amniotic fluid.

**Table 3 viruses-15-00819-t003:** Classification of prenatal cerebral ultrasound abnormalities, adapted from [159,167,168,169].

Prognostic	Cerebral Ultrasound Features
Mild and good prognosis *	ventriculomegaly <12 mm parenchymal calcifications subependymal cysts calcifications of lenticulostriate vessels periventricular hyper echogenicity
Severe	periventricular pseudo cysts (occipital/temporal horns) periventricular cystic leukomalacia severe ventriculomegaly >15 mm corpus callosum dysgenesis
Very severe	gyration anomaly (lissencephaly and polymicrogyria) microcephaly cerebellar hypoplasia

Note. * When any one of the above remains strictly isolated, the prognosis remains that of an asymptomatic neonate who has a higher risk of developing subsequent partial SNHL [170].

## Data Availability

Not applicable.

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
