# Peer review of "Primary, Secondary, and Tertiary Prevention of Congenital Cytomegalovirus Infection"

_viruses, 2023, doi:10.3390/v15040819_

Round 1

Reviewer 1 Report (New Reviewer)

This is an interesting and well-written review documenting the state of the field in congenital CMV infection.  The review is comprehensive and includes references to both earlier landmark studies and more current research, providing the reader with extensive knowledge of the subject.  Some sections are rather exhaustive and might benefit from fewer words and a graphic to illustrate.  Overall, the work represents a valuable contribution to the field and should be published.  Attention to some minor issues would improve the manuscript. The following are textual changes requiring clarification or correction:

Line 68-79    A broad claim about the most recent studies is made but only 1 study from France is cited here – revise language or add more references.

Line 75 persist all life – awkward wording -try “persists for life.”

Lines 97-102  this is very confusing and not clear what the point is  - high seroprevalence = lower congenital infection rates?  Please explain.

Line 108 awkward wording – consider” Knowledge and awareness among women”

Lines 174-195  The section appeared in red in my version as if it has been added.  It is noticeably dense and very wordy.  Can this section be streamlined?

Lines 197- 198  women expect to receive information on CMV from their health professional – are you speaking specifically about pregnant women or all women?  It might be better to say “women expect to receive information on CMV, along with other pregnancy-related risks, from their health professional”

Lines 206-207  Is the lack of knowledge because of poor coverage in the curriculum or because they don’t see it in clinic?   Please clarify or speculate.  This section is also in red like it was added later.

Line 271 references another review article.  What primary studies show the efficacy of NAb?

Line 283 – don’t end on a color, list the types of vaccine designs to be discussed

Figure 1 is missing the legend or explanation about pattern A and pattern B

Lines 483-485 says that 3 studies show efficacy of valganciclovir, but only one reference is cite -add 2 other refs

Line 497 – define QALY

Author Response

This is an interesting and well-written review documenting the state of the field in congenital CMV infection.  The review is comprehensive and includes references to both earlier landmark studies and more current research, providing the reader with extensive knowledge of the subject.  Some sections are rather exhaustive and might benefit from fewer words and a graphic to illustrate.  Overall, the work represents a valuable contribution to the field and should be published.  Attention to some minor issues would improve the manuscript. The following are textual changes requiring clarification or correction:

We thank the reviewer for this positive comment and for his careful reading of the manuscript. The comments made by the reviewer have been carefully studied and considered to make the necessary changes.

Line 68-79    A broad claim about the most recent studies is made but only 1 study from France is cited here – revise language or add more references.

  • Deleted for more clarity, and supplementary references added: However, European studies highlight the lack of knowledge and the frequent inaccuracy of the information given by health professionals to pregnant women”. Despite an evolution in virus awareness over the last decade, the most recent studies still show significant gaps in knowledge of the virus[21-23], which drastically reduces the effectiveness of primary prevention”. Lines 71-74

Line 75 persist all life – awkward wording -try “persists for life.”

  • Modified as required : “Like other DNA herpes viruses, after primary infection, CMV establishes as a latent viral infection that persists for life”. Lines 77-78

Lines 97-102  this is very confusing and not clear what the point is  - high seroprevalence = lower congenital infection rates?  Please explain.

  • Reworded for more clarity: “Non-primary infections may occur either due to an infection with a different strain (reinfection) or as a result of reactivation of the endogenous strain. The epidemiological importance of non-primary infections has been of increasing interest in recent years[35]. Recent studies have been able to highlight the importance of non-primary CMV infections[35-39], demonstrating, for example, that a country considered to have a high seroprevalence of CMV such as Brazil (>70% of the population), nevertheless reported a prevalence of CMV infection of about 1.1%[40]. Although the incidence of lesions due to cCMV infections is higher after a primary infection[36], these results allow to assume that non-primary infections can nevertheless be responsible for severe congenital infection of the newborn[35] just as it would seem that impacts on the newborn increase in presence of pre-existing maternal comorbidities such as hypertension or diabetes[36].” Lines 96-106

Line 108 awkward wording – consider” Knowledge and awareness among women”

  • Change for more clarity as: “The current literature agrees that increasing CMV knowledge and awareness among pregnant women, as well as its mode of transmission, seem to be effective to reduce rate of infection”. Lines 112-114

Lines 174-195  The section appeared in red in my version as if it has been added.  It is noticeably dense and very wordy.  Can this section be streamlined?

  • Reworded for more clarity: “The hypothesis developed several years ago suggesting a correlation between the ap-plication of hygiene measures and a low seroconversion rate[66,67] has since been supported by results of different studies[20,42,60]. In 2009 by Vauloup-Fellous et al. who found that careful application of hygiene measures could lead to a decrease in seroconversion rates up to 80% (N= 5/2583 (0.19%))[60], and more recently by Revello et al. (2015)[25] who also showed that awareness and knowledge of preventive hygiene measures, significantly reduced the rate of seroconversion in the sensitised group compared to the non-sensitised group (1.2% vs 7.6%; Δ = 6.4%; 95% CI 3.2–9.6; p-value <0.001). The authors were also able to determine that primary prevention decreased the rate of congenital infection in newborns (0.9% vs. 2.5% of cumulative incidences in the sensitised and non-sensitised groups respectively)[25]”. Lines 178-188

Lines 197- 198  women expect to receive information on CMV from their health professional – are you speaking specifically about pregnant women or all women?  It might be better to say “women expect to receive information on CMV, along with other pregnancy-related risks, from their health professional”

  • Changed as advised: “The role of the health professional in primary prevention is essential as women expect to receive information on CMV, along with other pregnancy-related risks, from their health professional”. Lines 201-203

Lines 206-207  Is the lack of knowledge because of poor coverage in the curriculum or because they don’t see it in clinic?   Please clarify or speculate.  This section is also in red like it was added later.

  • Further information added: “…it was shown that although there was a significant increase in medical students' awareness of CMV during the years of study, due to pre-clinical course on infectious disease given and the encounter with cCMV patients during clinical rotations, there was nevertheless a significant lack of knowledge regarding modes of transmission and available treatments at the end of the university curriculum[71]”. Lines 210-215

Line 271 references another review article.  What primary studies show the efficacy of NAb?

  • Reference changed as recommended : “It has been shown that NAbs play an important role not only in the prevention but also in the progression of disease[85]”. Lines 277-278

Line 283 – don’t end on a color, list the types of vaccine designs to be discussed

  • Types of vaccines are now listed: “Various strategies for developing an effective vaccine have been investigated by the scientific community in recent years. Vaccines currently under development are mainly based on the following designs: live-attenuated and disabled-infectious single cycle, adjuvanted recombinant protein, DNA, mRNA, virus-like particle, viral vectored, and peptide vaccines.” Lines 288-292

Figure 1 is missing the legend or explanation about pattern A and pattern B

  • We have added the following legend, from the original publication:“Relative changes in CMV IgM, IgG, and IgG avidity levels over time following primary CMV infection. IgM pattern A represents the typical IgM response pattern, whereas IgM pattern B represents long-term IgM persistence. In a CMV IgG-positive individual, an IgM-positive result of 20 indicates infection around 3 months previously if the individual exhibits IgM pattern A but around 6 months previously if the individual exhibits IgM pattern B. By employing CMV IgG avidity testing, the correct time since infection can be determined; a low-avidity result (expected to be about 30 based on this figure) indicates primary infection about 3 months previously, whereas a high-avidity result (expected to be about 70) indicates primary infection more than 6 months previously (Reuse of content from Prince et al., 2014 with permission[120])”. Lines 457-465

Lines 483-485 says that 3 studies show efficacy of valganciclovir, but only one reference is cite -add 2 other refs

  • We have added these references and specified that a meta-analysis of their results be conducted.: “Since then, a meta-analysis of the results of three recent studies has concluded that valaciclovir is effective in the secondary prevention of maternal-fetal transmission of CMV [110,137-139]”. Lines 522-524

Line 497 – define QALY

  • Defined as recommended : “An American cost-effectiveness analysis, based on 100,000$ per Quality-Adjusted Life Year (QALY) …” Line 524-525

Reviewer 2 Report (New Reviewer)

In “Primary, secondary, and tertiary prevention of congenital Cytomegalovirus infection” (Viruses manuscript # 2182445), Sartori et al. review a prevention strategy based upon 3 axes designed to reduce the risks of 1) CMV contamination, 2) materno-fetal transmission, and 3) symptomatic infections in newborns.  The goals of the review are to present currently available congenital CMV prevention strategies and assess their efficacy.  The review is written primarily from a public health perspective, covering epidemiological studies and some clinical studies.  The authors also discuss education/outreach approaches to reduce CMV transmission in women.  I found this public health approach to be the strength of the review.

The discussion of current vaccination strategies was a little problematic.  I have listed a few minor issues that should be addressed by the authors prior to publication.  These are mostly small oversights in the description of the CMV vaccines in development and clinical testing.  Otherwise, the review is clearly written and does a nice job of covering the relevant published papers.  I really enjoyed the overall layout of the review and think that it will be of broad general interest for research scientists focused on development of CMV vaccines and clinicians interested in reducing the incidence of congenital CMV.  It will be of specific interest for public health scientists focused on congenital CMV in addition to those interested in reducing disease burden in people with developing immune systems.

Minor Problems:

1.    Lines 310-337: Sections 2.4.3 and 2.4.4 (Both sections).  The authors do not describe the viral antigens that are encoded by the various DNA and mRNA vaccines that are currently in clinical trials.  This information is essential for a clear understanding of the current state of development of these vaccines and should be included.

2.    Lines 338-347: Section 2.4.5.  The authors do not describe the goal of vaccination with virus-like particles.  A short description of CMV glycoproteins, their role in viral entry, and neutralizing antibody responses elicited by these vaccines would provide sufficient background for the typical reader. These proteins include: gB, gH/gL, and the pentameric complex.

3.    Lines 348-361: Section 2.4.6.  The authors should include a brief description of the viral vectored vaccines in preclinical development targeting the pentameric complex as an additional strategy to gB vaccination.

4.    Line 375:  The abbreviation for “CoPs” should be described in full in its first usage.

5.    Line 770: I think the authors mean “sequelae”, not “sequels”

Author Response

In “Primary, secondary, and tertiary prevention of congenital Cytomegalovirus infection” (Viruses manuscript # 2182445), Sartori et al. review a prevention strategy based upon 3 axes designed to reduce the risks of 1) CMV contamination, 2) materno-fetal transmission, and 3) symptomatic infections in newborns.  The goals of the review are to present currently available congenital CMV prevention strategies and assess their efficacy.  The review is written primarily from a public health perspective, covering epidemiological studies and some clinical studies.  The authors also discuss education/outreach approaches to reduce CMV transmission in women.  I found this public health approach to be the strength of the review.

The discussion of current vaccination strategies was a little problematic.  I have listed a few minor issues that should be addressed by the authors prior to publication.  These are mostly small oversights in the description of the CMV vaccines in development and clinical testing.  Otherwise, the review is clearly written and does a nice job of covering the relevant published papers.  I really enjoyed the overall layout of the review and think that it will be of broad general interest for research scientists focused on development of CMV vaccines and clinicians interested in reducing the incidence of congenital CMV.  It will be of specific interest for public health scientists focused on congenital CMV in addition to those interested in reducing disease burden in people with developing immune systems.

We thank the reviewer for this very positive comment and for his help on the section on vaccination strategies. Changes requiring further analysis involving vaccines have been carefully considered to make the required modifications.

Minor Problems:

  1. Lines 310-337: Sections 2.4.3 and 2.4.4 (Both sections).  The authors do not describe the viral antigens that are encoded by the various DNA and mRNA vaccines that are currently in clinical trials.  This information is essential for a clear understanding of the current state of development of these vaccines and should be included.
  • Added as recommended: Section 2.4.3: “These DNA vaccines use the genes coding for the tegument protein pp65 and/or the envelop glycoprotein B (gB) [83]”. Lines 321-322

  • Added as recommended: Section 2.4.4: “These mRNA vaccines use the genes coding for the tegument protein pp65, gB and/or the pentamer complex (PC). It leads the immune system to produce responses against its corresponding antigen[80]. Currently, two of these vaccines are in Phase I, II or III testing[83]. ModernaTX, Inc. tested two mRNA vaccines; the mRNA-1647 vaccine (composed of one mRNA from gB and five mRNAs from PC) and the mRNA-1443 composed of pp65. They were evaluated in healthy adults in Phase I clinical trials. The mRNA-1647... “ Lines 333-339

  1. Lines 338-347: Section 2.4.5.  The authors do not describe the goal of vaccination with virus-like particles.  A short description of CMV glycoproteins, their role in viral entry, and neutralizing antibody responses elicited by these vaccines would provide sufficient background for the typical reader. These proteins include: gB, gH/gL, and the pentameric complex.
  • Completed as required: “The response of this vaccination would be the production of neutralizing antibodies against envelop glycoproteins like gB, gH/gL and PC. Indeed, these glycoproteins are involved in in attachment/entry of the virus in the human cell and neutralizing antibodies could prevent infection and spread to placental cells[94]”. Lines 355-359
  1. Lines 348-361: Section 2.4.6.  The authors should include a brief description of the viral vectored vaccines in preclinical development targeting the pentameric complex as an additional strategy to gB vaccination.
  • Completed as required: “In preclinical studies, viral vectored vaccines using envelop glycoproteins are promising targets. Both gB and PC vaccine candidates lead to neutralizing antibody response and a good strategy could be the association of several glycoproteins[95]. Outcome measures generally target T-cell responses, and CMV-specific T-cell antigens are usually included in the viral vector vaccine[82] “. Lines 372-376
  1. Line 375:  The abbreviation for “CoPs” should be described in full in its first usage.
  • Described previously in the text: “… but also that the immune correlates of protection (CoPs) are still unclear…” Lines 262-263
  1. Line 770: I think the authors mean “sequelae”, not “sequels”
  • Change as required: “Finally, to propose a curative treatment in infected fetuses presenting risks of adverse outcomes or sequelae based…”. Lines 796-798

Reviewer 3 Report (New Reviewer)

Sartori et al have gathered an impressive library of literature specifically on prevention of cCMV infection. This effort is an important contribution to the field.  However, the discussion overall has a bias toward European practices and many comments are not supported by the literature (see below). The manuscript would also benefit from significant review for English language, grammar, and terms. In particular, terms should be defined more clearly and used consistently through the paper, e.g. primary/non-primary infection should be defined early, “maternal infection” is not specific enough, and how primary/secondary/tertiary prevention are defined and apply to a setting where two entities are involved (e.g. secondary prevention for mother = primary prevention for fetus)

Section 1 Intro

Line 51-52

“infection” not “contamination”

Fetal not just symptomatic since we want to prevent all infections

Section 2 Primary prevention 

Lines 71-72

Awkward transition especially in a paragraph focusing on awareness – suggest delete

Line 83-84

“Daycare workers” not “daycare women”

Line 91-92

Use terms consistently, at least in the same sentence, e.g. here “prenatal” or “in utero” but not both.

Lines 93 and 95

Same for “secondary” and “non-primary”, pick one. The latter is used more commonly.

Line 108

Is there a difference between knowledge and awareness? If authors mean the same thing then pick one term for section title.

Section 2.2

Clarify if studies in this section examined all women or only pregnant women

Line 137-139

These numbers should be presented as estimates not exact

Line 150

Should read “not kissing”

Lines 160-163

Make sure “particular attention should be paid to this preventive measure when teaching pregnant women” is not the opinion of the authors but a conclusion of referenced studies.  If the former then should delete.

Line 174+

Seems odd that the first ref 60 is not discussed but the second ref 24 is discussed in detail. Consider putting the more detailed first or compare/contrast refs 24 and 60.

Line 191

Shouldn’t the word be “UNaware”?

Line 204

What types of healthcare professionals – med students but also OB or peds or other?

What audience – pregnant or all women?

Line 212

Do “diverse levels of knowledge gaps” differ by specialty or other variable?

Section 2.4 Vaccines

Suggest adding the vaccine development work of Don Diamond at City of Hope

Line 253

“Frequent re-infections and reactivations” are obstacles to vaccine efficacy, not development

Section 2.4.8

Suggest title “Correlates of Protection” or similar rather than uninformative “Strategy”

Section 3 secondary prevention 

Line 419-420

CMV IgG avidity testing usually does NOT confirm or exclude primary infection. There is a large gray area of intermediate results

Line 437

There is no way to confirm non-primary infection in a clinical setting (vs research setting) unless by chance there is a CMV IgG positive result in the record before pregnancy. (CMV neg result might be helpful if recent enough.)

Line 443-445

Measurement of CMV PCR in blood does NOT distinguish primary vs non-primary infection in any type of patent (immunocompromised or pregnant), so it is not a test used only to detect an “active non-primary infection”.

Line 451

I am not aware of any study linking reactivation in the kidneys (which is local) with fetal transmission, although it’s certainly associated with horizontal transmission. Make sure ref 121 actually makes this statement about kidneys.

Line 453

“and is thought to lead to maternal viremia” should read “..is more likely to lead to..” or similar

Line 454

Differentiating reactivation from reinfection is not only difficult, but it’s impossible in a clinical setting. Should clarify.

Line 468-477

This paragraph is inaccurate or at least misleading:

Actually the majority of specialists recommend against serologic screening (e.g. consensus guidelines Rawlinson 2017), although screening is routinely performed in some countries, regions, or local practices.

Screening may be of particular interest to women at risk, but that doesn’t mean it’s good clinical practice or standard of care.  Timing of primary infection using IgG avidity in the first trimester is often impossible because the results are unclear and/or don’t clarify whether maternal infection happened before or after conception.  This is an area of significant uncertainty in the field and should be presented as such.

Seroconversion can rarely be detected during the first trimester given lack of time or antecedent CMV IgG negative result, and the first trimester is too early for fetal diagnosis (either because too early after maternal primary infection or too early for imaging or amnio to be diagnostic – fetal blood sampling is too high risk).  The authors seem to state this later in lines 537-544.  The last sentence of this paragraph should be deleted.

This discussion should also mention availability of termination of pregnancy as a factor in the rationale for screening.  In this case, and if valacyclovir is also not readily available, then there are no options for interventions if screening suggests primary infection.

Line 499

Shouldn’t this read “..identification of maternal primary infection..”?

Section 3.2 title and line 537

Should read “…maternal primary infection”

Many other places in the paper when not clarified.

Lines 542-544

Amnio and CVS at what timepoints in pregnancy?

Lines 545+

Discussion of valacyclovir should highlight that this drug is not typically effective against CMV so has to be given in much higher doses than usual and therefore carries a higher risk of complications

Line 568

If simply the opinion of these authors, “probably due to lack of power” should be deleted

Line 570

Which study?

Lines 597-599

CMV HIg is NOT less-well tolerated than valacyclovir especially at high dose as above. This sentence should be deleted.

Lines 607-609

No we need LESS local policy- or setting-specific practices – we need more research to identify practice standards so patient care can be delivered consistently not just based on where a person lives.

Section 4 Tertiary prevention

How does tertiary prevention differ from routine clinical management (monitoring +/- treatment) of infected infants? This is a confusing term so should clarify distinction

Line 653

State that ultrasound refers to prenatal/fetal

Table 3

Where is this table from - ref 161 or the authors?

Ultrasound performed at what time? Text below says 3rd trimester

Does the list of features refer to specific criteria or just examples?

Line 674

Clarify that “..high sensitivity and specificity..” refers to fetal diagnosis by amnio, not to determining prognosis.

Section 4.2

Should not use “in utero treatment” in a section entitled “tertiary prevention” – use one term or the other

Lines 710-713

When was valacyclovir used in these studies.. 1st trimester?  If so then there is limited capacity to identify “mild to moderate signs” of cCMV and for the drug to have any effect on clinical disease during a stage of irreversible organogenesis especially the brain. This comment should be more clear about the lack of data on using this drug to “treat” fetuses in the first trimester.

Lines 715-718

Ganciclovir use during pregnancy remains limited due to teratogenicity – especially during the 1st trimester when fetal disease is usually the most severe (i.e. comparison to use in premature infants is not appropriate since the drug wouldn’t be used in the 3rd trimester) – not bone marrow toxicity, so this drug is NOT an alternative.

Lines 724-726

Monitoring of disease progression is crucial in all infected children not just asymptomatic, and does not belong in a section on prevention.

Lines 749-757

As above re section title, tertiary prevention just means clinical management. This is not a thorough discussion of clinical management and should be deleted

Author Response

Sartori et al. have gathered an impressive library of literature specifically on prevention of cCMV infection. This effort is an important contribution to the field.  However, the discussion overall has a bias toward European practices and many comments are not supported by the literature (see below). The manuscript would also benefit from significant review for English language, grammar, and terms. In particular, terms should be defined more clearly and used consistently through the paper, e.g. primary/non-primary infection should be defined early, “maternal infection” is not specific enough, and how primary/secondary/tertiary prevention are defined and apply to a setting where two entities are involved (e.g. secondary prevention for mother = primary prevention for fetus)

We thank the reviewer for his careful reading of the manuscript. We have tried in our revision to consider as much as possible the points raised by the different reviewers, whose opinion may differ. Overall, we have revised confusing points to make them more understandable.

We regret that the reviewer mentioned a bias towards European practices, while we used studies from North America, Asia, Australia or even Brazil contributing to the description of the burden of congenital CMV infections and the different prevention strategies. This impression could be more attributable to the fact we developed a lot on the strategies to be used in countries where half of congenital CMV infections are secondary to primary maternal infections (i.e. North America or Europe where seroprevalence in women of childbearing age is about 50%), and where there is an ongoing debate about screening for CMV during pregnancy and offering preventive therapies in case of infection. Of course, this strategy to reduce the burden of congenital CMV may be less effective in countries with high seroprevalence (such as Brazil with >70% seroprevalence) and where congenital CMV infections are more often consecutive to secondary infections. Throughout the review, we have tried to clarify the context of the different strategies proposed to avoid this bias.

As specified early in the review, “maternal infection” refers to primary and non-primary CMV infection in the mother. To avoid any misunderstanding, we have specified:

  • “2.1. Primary and non-primary maternal infections” Line 64-65

Primary, secondary and tertiary preventions in the context of pregnancy are defined as interventions used to avoid maternal exposure, to reduce fetal exposure and to improve fetal or neonatal outcomes after exposure, respectively. These definitions are adapted from the operational definition of the WHO to the context of maternal-fetal infections[7-9] but also in other contexts such as prematurity [10]. These definitions are presented in the introduction, and are clarified at the beginning of each section:

  • Introduction “To reduce the likelihood of maternal infection, maternal-fetal transmission, and subsequent complications in the event of congenital infection, the prevention strategy is structured around three key components: primary, secondary, and tertiary prevention [14-17].” Lines 54-56

  • Primary prevention: to reduce the risk of maternal primary infection and re-infection. “Primary prevention consists of coordinated actions to prevent foreseeable problems, protect existing states of health and healthy functioning, and promote the potentials of individuals and groups in their physical and sociocultural environments over time[19]. Knowledge of CMV and its mode of transmission is the cornerstone of primary prevention, enabling women of childbearing age and in early pregnancy to apply adapted hygienic measures to avoid CMV primary infection or re-infection[20]”. Lines 66-71

  • Secondary prevention: to reduce the risk of maternal-fetal transmission. “Secondary prevention aims to reduce the impact of a disease that has already occurred. It consists of detecting and treating the disease as early as possible to stop or slow down its progression and prevent long-term problems[104]. Secondary prevention aims to avoid or reduce the risk of maternal-fetal transmission in case of proven CMV seroconversion in the mother[105]. Literature on secondary prevention of cCMV focuses on two complementary topics: maternal serological screening[106,107] and prophylactic therapies including administration of hyperimmune globulin[24,108,109] or antiviral therapy[110]”. Lines 419-426

  • Tertiary prevention: to reduce the risk of symptomatic cCMV infections. “Finally, tertiary prevention aims to lower the impact of a disease that has long term effects. It involves developing follow-up or treatment to improve as much as possible the ability to function, the quality of life and life expectancy[104]. Tertiary prevention for cCMV is mainly based on surveillance for signs of fetal infection caused by the virus[7]. In the case of prenatal diagnosis of congenital infection, recent advances in fetal medicine have made it possible to propose a prognosis work-up based on ultrasound, MRI [155-157], fetal blood sampling[158], and a curative in-utero treatment for mild to moderate forms of disease.” Lines 665-672

Section 1 Intro

Line 51-52

“infection” not “contamination”

Fetal not just symptomatic since we want to prevent all infections

  • Change as required: “To reduce the likelihood of maternal infection, maternal-fetal transmission, and subsequent complications in the event of congenital infection, the prevention strategy is structured around three key components: primary, secondary, and tertiary prevention”. Lines 54-55

Section 2 Primary prevention 

Lines 71-72

Awkward transition especially in a paragraph focusing on awareness – suggest delete

  • Deleted as required for more clarity: However, European studies highlight the lack of knowledge and the frequent inaccuracy of the information given by health professionals to pregnant women”. Line 71

Line 83-84

“Daycare workers” not “daycare women”

  • Changed as required: “…identified that daycare worker and pregnant women…” Lines 86-87

Line 91-92

Use terms consistently, at least in the same sentence, e.g. here “prenatal” or “in utero” but not both.

  • Changed as required: “…only prenatal transmissions have been correlated with cCMV infection[29,33,34]”. Lines 94-95

Lines 93 and 95

Same for “secondary” and “non-primary”, pick one. The latter is used more commonly.

  • Changed as required; Secondary infections => non-primary infections Lines; 96,99,103-104,540

Line 108

Is there a difference between knowledge and awareness? If authors mean the same thing then pick one term for section title.

  • Knowledge and awareness are distinct concepts that each bring a unique perspective to the subject of interest:
    • Awareness is perceiving, knowing, feeling, or being conscious of events, objects, thoughts, emotions, or sensory patterns.
    • Knowledge is facts, information, and skills acquired through experience or education, the theoretical or practical understanding of a subject.

Section 2.2

Clarify if studies in this section examined all women or only pregnant women

  • Clarified as recommended: “A low level of CMV-related knowledge was also found among woman of childbearing age and/or pregnant woman in other studies[43,45-48]. This rate varies greatly from one country to another with a percentage of knowledge of less than 20% reported in some studies conducted in Ireland, the USA, Holland, Japan, Saudi Arabia and Australia[32,45-47,49-51]. Three studies conducted in Canada, the USA and England be-tween 2019 and 2021 have shown a rate of CMV awareness among pregnant women of between 32.4% and 39%[42,52,53], while three others conducted in Switzerland, Italy and France showed awareness of pregnant women rates ranging from 39% to 60%[15,54,55]”. Lines 121-129

Line 137-139

These numbers should be presented as estimates not exact

  • Clarified as required: “In the United States, the number of children born with cCMV and long-term sequelae is estimated at 8,600 per year, which is much higher than the number of children born over the same period with sequelae due to toxoplasmosis and Down syndrome (about 1'000 and 4'000 cases, respectively), diseases yet better known by women[57].” Lines 140-144.

Line 150

Should read “not kissing”

  • Changed as recommended: …not kissing on the lips.. Line 154

Lines 160-163

Make sure “particular attention should be paid to this preventive measure when teaching pregnant women” is not the opinion of the authors but a conclusion of referenced studies.  If the former then should delete.

  • Reformulated for more neutrality: “Ease of implementation of the preventive measure agreeing not to kiss infants on the lips appears to be more difficult for respondents to put into practice[53]. As this habits can be cultural and subject to demographic variables, particular attention should be paid to this socio-cultural aspect when teaching pregnant women, to ensure that the preventive message will be optimized and delivered appropriately[32]”. Lines 163-167

Line 174+

Seems odd that the first ref 60 is not discussed but the second ref 24 is discussed in detail. Consider putting the more detailed first or compare/contrast refs 24 and 60.

  • Reworded to emphasize the temporality of the studies rather than the comparison between the two studies: “The hypothesis developed several years ago suggesting a correlation between the application of hygiene measures and a low seroconversion rate[66,67] has since been supported by results of different studies[20,42,60]. In 2009 by Vauloup-Fellous et al. who found that careful application of hygiene measures could lead to a decrease in seroconversion rates up to 80% (N= 5/2583 (0.19%))[60], and more recently by Revello et al. (2015)[25] who also showed that awareness and knowledge of preventive hygiene measures, significantly reduced the rate of seroconversion in the sensitised group compared to the non-sensitised group (1.2% vs 7.6%; Δ = 6.4%; 95% CI 3.2–9.6; p <0.001). The authors were also able to determine that primary prevention decreased the rate of congenital infection in newborns (0.9% vs. 2.5% of cumulative incidences in the sensitised and non-sensitised groups respectively)[25]”. Lines 178-188

Line 191

Shouldn’t the word be “UNaware”?

  • Reworded to avoid confusion: “Despite this, it has been shown that a large majority were informed about CMV and preventive hygiene measures at their first consultation, often at the end of the first trimester[54]”. Lines 193-196

Line 204

What types of healthcare professionals – med students but also OB or peds or other?

Do “diverse levels of knowledge gaps” differ by specialty or other variable?

  • The medical students recruited in this study were in their first four years of training before residency, not permitting to differentiate the knowledge of future obstetricians from that of future pediatricians or future GP. However, second, third- and fourth-year students seem to have better knowledge than first year students. Specified for clarity: “Using a questionnaire to assess their general knowledge of CMV and cCMV during their first four years of training, it was shown that although there was a significant increase in medical students' awareness of CMV during the years of study, due to the pre-clinical infectious disease course given at the end of the first year and the encounter with cCMV patients during clinical rotations, there was nevertheless a significant lack of knowledge regarding modes of transmission and available treatments at the end of the university curriculum[71]”. Lines 209-215

Section 2.4 Vaccines

Suggest adding the vaccine development work of Don Diamond at City of Hope

  • Integrated reference as suggested: “Recently, in phase I clinical trials, one of them (CMV-MVA triplex vaccine of City of Hope Medical Center) have not only demonstrated a safe profile among participants HSCT treated with the vaccine, but also the ability to elicit robust cytotoxic T lymphocyte (CTL) responses through donor vaccination, in CMV-positive recipients[83,96]. Indeed, CMV‐specific CD137+CD8+ T cells were significantly higher (p-value < 0.0001 and p-value = 0.0174 respectively) in recipients of Triplex vaccinated matched related donor than unvaccinated ones (control cohort)[96]”. Lines 376-383

Line 253

“Frequent re-infections and reactivations” are obstacles to vaccine efficacy, not development

  • Changed as required: “Frequent reinfections and reactivations of the virus are the main obstacles to vaccine efficacity”. Lines 259-260

Section 2.4.8

Suggest title “Correlates of Protection” or similar rather than uninformative “Strategy”

  • Changed as required: Correlates of Protection Line 396

Section 3 secondary prevention 

Line 419-420

CMV IgG avidity testing usually does NOT confirm or exclude primary infection. There is a large gray area of intermediate results

  • Weighted paragraph as recommended: “Furthermore, although low CMV IgG avidity confirms a recent primary infection while high CMV IgG avidity excludes it, there is a large gray area in interpreting results where intermediate IgG avidity cannot fully exclude a recent primary infection[114]”. Lines 443-446

Line 437

There is no way to confirm non-primary infection in a clinical setting (vs research setting) unless by chance there is a CMV IgG positive result in the record before pregnancy. (CMV neg result might be helpful if recent enough.)

  • We agree with this comment, and the references cited, particularly the recent study by Perillaud-Dubois and colleagues [11] indicate that even with positive IgG before pregnancy, most patients with non-primary CMV infection during pregnancy do not have an increase in IgG titer. We have clarified to avoid any misunderstanding: “However, serological screening is only reliable to detect primary infections. In most patients with confirmed CMV non-primary infection during pregnancy (positive IgG before pregnancy with re-activation of viremia during pregnancy and / or cCMV infection in the newborn), serology fails in detecting CMV re-infection or reactivation [36,37,121,122]. Indeed, >50% women immunized before pregnancy and delivering an infected baby have stable CMV-IgG titers, negative CMV-IgM and negative PCR in se-rum. Moreover, increased CMV-IgG titration, as well as a high CMV-IgG avidity index, and/or a positive CMV-IgM can be attributed to other clinical situations more frequently encountered than CMV non-primary infection”. Lines 470-478

Line 443-445

Measurement of CMV PCR in blood does NOT distinguish primary vs non-primary infection in any type of patent (immunocompromised or pregnant), so it is not a test used only to detect an “active non-primary infection”.

  • We agree that a positive CMV PCR in whole blood does not distinguish between a primary and a non-primary infection without the patient's history. However, in a pregnant woman with documented pre-pregnancy immunity, the viremia of the primary infection is not expected to persist up to the current pregnancy. Thus, we clarified that only a positive CMV PCR in pregnant women with positive IgG before pregnancy could indicate an active non-primary infection: “Detection of CMV-DNA by PCR in whole blood can indicate both primary and non-primary CMV infection [123]. Viremia in non-primary infection during pregnancy, as in immunocompetent non-pregnant individuals, seems to be transient and the viral load can be very low, limiting its detection[124]. However, when CMV-DNA is detected by PCR in a pregnant woman with positive IgG prior to pregnancy, this could indicate non-primary infection. But, when CMV-DNA is not detected by PCR, it does not exclude non-primary infection.” Lines 479-484

Line 451

I am not aware of any study linking reactivation in the kidneys (which is local) with fetal transmission, although it’s certainly associated with horizontal transmission. Make sure ref 121 actually makes this statement about kidneys.

  • Deleted as required: “In this situation, reactivation in the macrophages of the uterus[125], and in the cervix could be responsible for fetal infection, without detectable viremia in the mother[126].”. Line 485-487

Line 453

“…and is thought to lead to maternal viremia” should read “..is more likely to lead to..” or similar

  • Changed as required: “Reinfection, on the other hand, is an infection with a new viral strain and is it more likely to lead to maternal viremia”. Lines 487-488

Line 454

Differentiating reactivation from reinfection is not only difficult, but it’s impossible in a clinical setting. Should clarify.

  • Changed as required: “Differentiate reactivation from reinfection is impossible using standard serologies. However, the appearance of new antibody specificity against polymorphic epitopes of CMV, detected by strain-specific ELISA, may indicate reinfection[127,128]. Diagnosis of non-primary infection during pregnancy remains challenging and explains why routine molecular serologic analysis is not considered in this context[37]”. Lines 488-493

Line 468-477

This paragraph is inaccurate or at least misleading:

Actually the majority of specialists recommend against serologic screening (e.g. consensus guidelines Rawlinson 2017), although screening is routinely performed in some countries, regions, or local practices.

Screening may be of particular interest to women at risk, but that doesn’t mean it’s good clinical practice or standard of care.  Timing of primary infection using IgG avidity in the first trimester is often impossible because the results are unclear and/or don’t clarify whether maternal infection happened before or after conception.  This is an area of significant uncertainty in the field and should be presented as such.

Seroconversion can rarely be detected during the first trimester given lack of time or antecedent CMV IgG negative result, and the first trimester is too early for fetal diagnosis (either because too early after maternal primary infection or too early for imaging or amnio to be diagnostic – fetal blood sampling is too high risk).  The authors seem to state this later in lines 537-544.  The last sentence of this paragraph should be deleted.

This discussion should also mention availability of termination of pregnancy as a factor in the rationale for screening.  In this case, and if valacyclovir is also not readily available, then there are no options for interventions if screening suggests primary infection.

The statement that the majority of specialists advise against serological screening may have been true 6 years ago, but it seems outdated now[12]. It depends mainly on the seroprevalence of CMV in women of childbearing age and the availability of screening, prenatal diagnosis, treatment and termination of pregnancy at reasonable costs in the countries concerned. Thus, it does not seem reasonable that this recommendation is universal, but rather based on the context of each country.

To answer point by point to this comment:

  • Pregnant women at risk are mainly those who were seronegative at their previous pregnancy, as they present a 24- and 6-fold higher risks of having a child with cCMV infection and sequelae related to this infection than the general pregnant population, respectively [13]. Therefore, to establish if a pregnant woman is at risk and should benefit from screening in the current pregnancy is mainly based on the screening results of her previous pregnancy.
  • Timing of primary infection: low CMV IgG avidity indicates that the primary infection occurred in the last 3 months and a high avidity excludes a primary infection in the last 3 months. The grey zone concerns cases with avidity classified as “intermediate” in which a primary infection in the last 3 months cannot be ruled out. For these cases, additional assays and follow-up of IgG, IgM and avidity kinetics can help to estimate the time of primary infection[14]. Algorithms using IgM and IgG detection and avidity combined with high throughput automated platforms have proven to be very sensitive in diagnosing and estimating the onset of primary maternal infection [15-19]. The main pitfall remains the interpretation of virological profile in serum with low level of IgG. Overall, the question should not be “Should we perform serology during pregnancy in places where seroprevalence warrants it and where it is available and reliable at a reasonable price?” but “When should we perform this serology in these settings”?

Indeed, as raised by the reviewer, the difficulty in interpreting avidity lies in differentiating pre-conception infections from infections during pregnancy. This is true if serology is performed at 12 weeks of gestation or later, but we have known for a decade that results from serology performed at 8 weeks are much easier to interpret [20]. Thus, the proposed approach to routine screening, in health systems that can accommodate it, is to perform a first serology in early pregnancy, at 6-8 w, and a second at the end of the first trimester.

  • Fetal diagnosis and prognosis: Fetuses that will develop the most severe forms of congenital infection are those infected after maternal infection in the periconceptional period or in the first trimester. Diagnosis of congenital infection can be reliably made by amniocentesis from 17 w. The prognosis of infected fetuses will be reliably obtained by a combination of ultrasound and fetal blood sampling in the 2nd trimester, or a series of ultrasounds and fetal brain MRI in the 3rd trimester, both of which have a negative predictive value of >95% for symptomatic cCMV infections [21]. This means that depending on the laws of each country on termination of pregnancy and the risks that pregnant women are willing to take (FBS are associated with a 2% risk of fetal loss), a reliable prognosis following diagnosis of congenital infection can be obtained before 24 w or before 32 w.
  • Even if termination of pregnancy is not an option in every country or is not considered an option for some pregnant women, having an antiviral treatment that reduces the rate of maternal-fetal transmission by 3 [22] and the rate of sequelae in case of fetal infection by 2 [23] remains a strategy that ethically justifies information on the possibility of serology at the beginning of pregnancy, prenatal diagnosis and dual-action prenatal treatment in case of maternal infection [24].
  • This strategy could be cost-effective in several European countries and in Japan, depending on the national costs of care.

To reflect this comment, we have revised this section to highlight the context in which routine screening could be considered: “Although various reasons have been advanced for questioning the relevance of routine serological screening of pregnant women (difficulty of interpretation, lack of treatment available...)[4], some specialists advocate for informing all pregnant women about the possibility of serological screening in early pregnancy, in order to optimise the detection and follow-up of congenital infections[112,134]. If this screening is desired by the pregnant woman and/or locally recommended, it should be offered as early as possible during pregnancy, and ideally pre-conceptionally to simplify the interpretation of results. Nevertheless, it is important to note that the discussion on serological screening in early pregnancy must consider the availability of screening, prenatal diagnosis, treatment and termination of pregnancy at reasonable costs in the countries concerned, and that the overall value of such screening also depends on the local epidemiology of CMV. Thus, it seems difficult to propose universal recommendations for CMV screening in pregnancy. “ Lines 503-514

Line 499

Shouldn’t this read “..identification of maternal primary infection..”?

  • Changed as required: “The different screening strategies can only identify maternal primary infections…” Lines 537

Section 3.2 title and line 537

Should read “…maternal primary infection”

Many other places in the paper when not clarified.

  • - Modified (also in other sections of the manuscript) : “In the case of proven maternal primary infection…” Line 577

Lines 542-544

Amnio and CVS at what timepoints in pregnancy?

  • Already stated earlier, but added as required: ”In the case of proven maternal primary infection, cCMV infection can be diagnosed by amniocentesis from 17 weeks gestation or possibly by chorionic villus sampling performed at 11-14 weeks' gestation. In order to achieve optimal sensitivity of these diagnostic tests (>95%), a consensus time frame of 6 to 8 weeks after primary infection should be respected (unless there are ultrasound signs).” Lines 577-580

Lines 545+

Discussion of valacyclovir should highlight that this drug is not typically effective against CMV so has to be given in much higher doses than usual and therefore carries a higher risk of complications

  • This discussion has been added: “It should be noted that the dosage of Valaciclovir in these three studies is 4 times higher than that used for its usual use (the dosage of 8g/d comes from studies showing its effectiveness on CMV reactivation and reinfection in renal transplant patients [149]). Side effects for pregnant women could be more frequent with this dosage. For example, among these three trials, two cases of acute renal failure (approximately 1% of patients) were identified after initiation of valaciclovir (8g per day) and resolved spontaneously after discontinuation of treatment [137]. These data highlight the importance of clinical and laboratory monitoring throughout the duration of treatment. For the moment, none of the data from these three studies is alarming for exposed fetuses, but it remains important to monitor them over the long term through pharmaco-epidemiological studies in order to ensure the safety of this dosage for the fetus”. Lines 618-627

Line 568

If simply the opinion of these authors, “probably due to lack of power” should be deleted

  • Deleted as required: “…(statistically not significant, probably due to lack of power). Line 609

Line 570

Which study?

  • Changed for more details: “Furthermore, Egloff et al. (2022) showed that valaciclovir was more effective in patients with positive maternal viremia (assessed by PCR in maternal blood)[26]”. Lines 612-614

Lines 597-599

CMV HIg is NOT less-well tolerated than valacyclovir especially at high dose as above. This sentence should be deleted.

  • This sentence has been modified to read: “However, the fact remains that hyperimmune globulin therapy during pregnancy is much more expensive than valaciclovir, and may cause severe allergic reactions in rare cases (1/203 in the study by Hughes et al. [111]) “. Lines 648-651

Lines 607-609

No we need LESS local policy- or setting-specific practices – we need more research to identify practice standards so patient care can be delivered consistently not just based on where a person lives.

  • This request for change is contrary to the reviewer's general comment, and to his earlier requests, to base our review less on bias towards “European practices”, and to include a more general review. This paragraph shows that the strategy to be favored between primary prevention and routine screening with the possibility of secondary treatment depends on the local epidemiology and the possibilities of each country for the management of CMV during pregnancy. We therefore propose to maintain this paragraph.

Section 4 Tertiary prevention

How does tertiary prevention differ from routine clinical management (monitoring +/- treatment) of infected infants? This is a confusing term so should clarify distinction

  • The definition of primary, secondary and tertiary prevention has been discussed above in our rebuttal. Tertiary prevention of maternal-fetal infection during pregnancy does not modify the routine clinical management of infected infants, who will be monitored by clinical, laboratory and imaging tests and treated with Ganciclovir or Valganciclovir if they meet the criteria for treatment. However, tertiary prevention in pregnancy does impact the management of infected fetuses, as it provides pregnant women with the opportunity to receive a prognosis based on ultrasound, MRI, and fetal blood sampling for their infected fetus, to receive treatment aimed at reducing the risk of sequelae in fetuses exhibiting mild to moderate signs of cCMV infection, and to be informed of the option of terminating the pregnancy if the fetus has severe brain abnormalities with a high risk of sequelae, in accordance with local laws. We have clarified this term at the beginning of the section: “Tertiary prevention for cCMV is mainly based on surveillance for signs of fetal infection caused by the virus[7]. In the case of prenatal diagnosis of cCMV infection, recent advances in fetal medicine have made it possible to propose a prognosis work-up based on ultrasound, MRI [156-158], and fetal blood sampling[159]; and a curative in-utero treatment reducing the risk of sequelae in fetuses exhibiting mild to moderate signs of cCMV infection.” Lines 665-670
  •  

Line 653

State that ultrasound refers to prenatal/fetal

  • Modified to read: “Prenatal ultrasound features can be labeled as extracerebral and cerebral findings.” Line 693

Table 3

Where is this table from - ref 161 or the authors?

Ultrasound performed at what time? Text below says 3rd trimester

Does the list of features refer to specific criteria or just examples?

  • This table is adapted from ref 156,164,165, as stated above the table, and can be used at any stage of pregnancy after 20 weeks, except for polymicrogyria which will be mainly diagnosed in the third trimester. We have modified to clarify: “Classification of prenatal cerebral ultrasound abnormalities, adapted from[156,164,165] ”. Line 709

Line 674

Clarify that “..high sensitivity and specificity..” refers to fetal diagnosis by amnio, not to determining prognosis.

  • We have deleted this sentence to avoid misunderstanding.

Section 4.2

Should not use “in utero treatment” in a section entitled “tertiary prevention” – use one term or the other

  • We have modified to read: “2. Preventing the risk of sequelae in fetuses with mild to moderate cCMV infection”

Lines 710-713

When was valacyclovir used in these studies.. 1st trimester?  If so then there is limited capacity to identify “mild to moderate signs” of cCMV and for the drug to have any effect on clinical disease during a stage of irreversible organogenesis especially the brain. This comment should be clearer about the lack of data on using this drug to “treat” fetuses in the first trimester.

  • Valaciclovir has been used from 20w in these studies, after amniocentesis to confirm fetal infection. The effect of valaciclovir at this stage of pregnancy is not to prevent abnormal neurogenesis, but to prevent an acute cerebral inflammatory response and encephalitis in the infected fetus. We have modified to clarify: “Fetuses with positive PCR on amniotic fluid and mild to moderate signs on mid-trimester or third-trimester ultrasound can be treated with transplacental Valaciclovir to reduce viral replication in the fetoplacental compartment[162,176,177], which could lead to improved perinatal and long-term outcomes, according to a phase II observational study[177].” Lines 763-767

Lines 715-718

Ganciclovir use during pregnancy remains limited due to teratogenicity – especially during the 1st trimester when fetal disease is usually the most severe (i.e. comparison to use in premature infants is not appropriate since the drug wouldn’t be used in the 3rd trimester) – not bone marrow toxicity, so this drug is NOT an alternative.

  • This section is about tertiary prevention, i.e. therapies used after the diagnosis of a fetal infection in the second or third trimester of pregnancy. Valganciclovir is of course not used during the first trimester of pregnancy dur to teratogenicity, but the case reports mentioned in this review, and preliminary results of a cohort study from Necker Hospital (presented by Y. Ville in the ISUOG world congress 2022) are encouraging to improve the outcome of infected fetuses presenting moderate to severe signs in the second or third trimester. However, in response to this comment, we have modified the text to avoid any misunderstanding: “The recommended treatment for infected and symptomatic neonates is Valganciclovir, which rapidly achieves negative viral loads and improve hearing and developmental outcomes[178], but its utilization during pregnancy remains limited due to concerns about teratogenicity and bone marrow toxicity[179]. However, in view of its ability to cross the placental barrier[180] and its use in hypotrophic and premature infants[181], its use during the second and third trimesters of pregnancy could be an alternative to Valaciclovir. Moreover, some reports[151,182] of Valganciclovir treatment of pregnant women with cCMV infection seem to support its safety and efficacy during the second and third trimesters of pregnancy.” Lines 767-775

Lines 724-726

Monitoring of disease progression is crucial in all infected children not just asymptomatic, and does not belong in a section on prevention.

Lines 749-757

As above re section title, tertiary prevention just means clinical management. This is not a thorough discussion of clinical management and should be deleted

  • To fulfil this comment, we have deleted this last section.

Bibliography

  1. Adler, S.P.; Finney, J.W.; Manganello, A.M.; Best, A.M. Prevention of child-to-mother transmission of cytomegalovirus by changing behaviors: a randomized controlled trial. Pediatr Infect Dis J 1996, 15, 240-246, doi:10.1097/00006454-199603000-00013.
  2. Adler, S.P.; Finney, J.W.; Manganello, A.M.; Best, A.M. Prevention of child-to-mother transmission of cytomegalovirus among pregnant women. J Pediatr 2004, 145, 485-491, doi:10.1016/j.jpeds.2004.05.041.
  3. Vauloup-Fellous, C.; Picone, O.; Cordier, A.G.; Parent-du-Chatelet, I.; Senat, M.V.; Frydman, R.; Grangeot-Keros, L. Does hygiene counseling have an impact on the rate of CMV primary infection during pregnancy? Results of a 3-year prospective study in a French hospital. J Clin Virol 2009, 46 Suppl 4, S49-53, doi:10.1016/j.jcv.2009.09.003.
  4. Billette de Villemeur, A.; Tattevin, P.; Salmi, L.R.; French Haut Conseil de la sante publique Working, G. Hygiene promotion might be better than serological screening to deal with Cytomegalovirus infection during pregnancy: a methodological appraisal and decision analysis. BMC Infect Dis 2020, 20, 418, doi:10.1186/s12879-020-05139-8.
  5. Calvert, A.; Vandrevala, T.; Parsons, R.; Barber, V.; Book, A.; Book, G.; Carrington, D.; Greening, V.; Griffiths, P.; Hake, D.; et al. Changing knowledge, attitudes and behaviours towards cytomegalovirus in pregnancy through film-based antenatal education: a feasibility randomised controlled trial of a digital educational intervention. BMC Pregnancy Childbirth 2021, 21, 565, doi:10.1186/s12884-021-03979-z.
  6. Revello, M.G.; Tibaldi, C.; Masuelli, G.; Frisina, V.; Sacchi, A.; Furione, M.; Arossa, A.; Spinillo, A.; Klersy, C.; Ceccarelli, M.; et al. Prevention of Primary Cytomegalovirus Infection in Pregnancy. EBioMedicine 2015, 2, 1205-1210, doi:10.1016/j.ebiom.2015.08.003.
  7. Nyholm, J.L.; Schleiss, M.R. Prevention of maternal cytomegalovirus infection: current status and future prospects. Int J Womens Health 2010, 2, 23-35, doi:10.2147/ijwh.s5782.
  8. Walker, S.P.; Palma-Dias, R.; Wood, E.M.; Shekleton, P.; Giles, M.L. Cytomegalovirus in pregnancy: to screen or not to screen. BMC Pregnancy Childbirth 2013, 13, 96, doi:10.1186/1471-2393-13-96.
  9. Mazzitelli, M.; Micieli, M.; Votino, C.; Visconti, F.; Quaresima, P.; Strazzulla, A.; Torti, C.; Zullo, F. Knowledge of Human Cytomegalovirus Infection and Prevention in Pregnant Women: A Baseline, Operational Survey. Infect Dis Obstet Gynecol 2017, 2017, 5495927, doi:10.1155/2017/5495927.
  10. Iams, J.D.; Romero, R.; Culhane, J.F.; Goldenberg, R.L. Primary, secondary, and tertiary interventions to reduce the morbidity and mortality of preterm birth. Lancet 2008, 371, 164-175, doi:10.1016/S0140-6736(08)60108-7.
  11. Perillaud-Dubois, C.; Letamendia, E.; Bouthry, E.; Rafek, R.; Thouard, I.; Vieux-Combe, C.; Picone, O.; Cordier, A.G.; Vauloup-Fellous, C. Cytomegalovirus Specific Serological and Molecular Markers in a Series of Pregnant Women With Cytomegalovirus Non Primary Infection. Viruses 2022, 14, doi:10.3390/v14112425.
  12. Ville, Y. Advocating for cytomegalovirus maternal serologic screening in the first trimester of pregnancy: if you do not know where you are going, you will wind up somewhere else. Am J Obstet Gynecol MFM 2021, 3, 100356, doi:10.1016/j.ajogmf.2021.100356.
  13. M. Leruez-Ville, Y.V. Épidémiologie et diagnostic virologique de l’infection congénitale à cytomégalovirus (CMV). Elsevier Masson SAS 2020, 204, 126-136, doi:https://doi.org/10.1016/j.banm.2019.10.020.
  14. Muller, J.; Flindt, J.; Pollmann, M.; Saschenbrecker, S.; Borchardt-Loholter, V.; Warnecke, J.M. Efficiency of CMV serodiagnosis during pregnancy in daily laboratory routine. J Virol Methods 2023, 314, 114685, doi:10.1016/j.jviromet.2023.114685.
  15. Revello, M.G.; Fabbri, E.; Furione, M.; Zavattoni, M.; Lilleri, D.; Tassis, B.; Quarenghi, A.; Cena, C.; Arossa, A.; Montanari, L.; et al. Role of prenatal diagnosis and counseling in the management of 735 pregnancies complicated by primary human cytomegalovirus infection: a 20-year experience. J Clin Virol 2011, 50, 303-307, doi:10.1016/j.jcv.2010.12.012.
  16. Lagrou, K.; Bodeus, M.; Van Ranst, M.; Goubau, P. Evaluation of the new architect cytomegalovirus immunoglobulin M (IgM), IgG, and IgG avidity assays. J Clin Microbiol 2009, 47, 1695-1699, doi:10.1128/JCM.02172-08.
  17. Delforge, M.L.; Desomberg, L.; Montesinos, I. Evaluation of the new LIAISON((R)) CMV IgG, IgM and IgG Avidity II assays. J Clin Virol 2015, 72, 42-45, doi:10.1016/j.jcv.2015.09.002.
  18. Chiereghin, A.; Pavia, C.; Gabrielli, L.; Piccirilli, G.; Squarzoni, D.; Turello, G.; Gibertoni, D.; Simonazzi, G.; Capretti, M.G.; Lanari, M.; et al. Clinical evaluation of the new Roche platform of serological and molecular cytomegalovirus-specific assays in the diagnosis and prognosis of congenital cytomegalovirus infection. J Virol Methods 2017, 248, 250-254, doi:10.1016/j.jviromet.2017.08.004.
  19. Sellier, Y.; Guilleminot, T.; Ville, Y.; Leruez-Ville, M. Comparison of the LIAISON((R)) CMV IgG Avidity II and the VIDAS((R)) CMV IgG Avidity II assays for the diagnosis of primary infection in pregnant women. J Clin Virol 2015, 72, 46-48, doi:10.1016/j.jcv.2015.08.018.
  20. Prince, H.E.; Lape-Nixon, M. Role of cytomegalovirus (CMV) IgG avidity testing in diagnosing primary CMV infection during pregnancy. Clin Vaccine Immunol 2014, 21, 1377-1384, doi:10.1128/CVI.00487-14.
  21. Leruez-Ville, M.; Ghout, I.; Bussieres, L.; Stirnemann, J.; Magny, J.F.; Couderc, S.; Salomon, L.J.; Guilleminot, T.; Aegerter, P.; Benoist, G.; et al. In utero treatment of congenital cytomegalovirus infection with valacyclovir in a multicenter, open-label, phase II study. Am J Obstet Gynecol 2016, 215, 462 e461-462 e410, doi:10.1016/j.ajog.2016.04.003.
  22. Shahar-Nissan, K.; Pardo, J.; Peled, O.; Krause, I.; Bilavsky, E.; Wiznitzer, A.; Hadar, E.; Amir, J. Valaciclovir to prevent vertical transmission of cytomegalovirus after maternal primary infection during pregnancy: a randomised, double-blind, placebo-controlled trial. Lancet 2020, 396, 779-785, doi:10.1016/S0140-6736(20)31868-7.
  23. Leruez-Ville, M.; Stirnemann, J.; Sellier, Y.; Guilleminot, T.; Dejean, A.; Magny, J.F.; Couderc, S.; Jacquemard, F.; Ville, Y. Feasibility of predicting the outcome of fetal infection with cytomegalovirus at the time of prenatal diagnosis. Am J Obstet Gynecol 2016, 215, 342 e341-349, doi:10.1016/j.ajog.2016.03.052.
  24. D'Antonio, F.; Marinceu, D.; Prasad, S.; Khalil, A. Effectiveness and safety of prenatal valacyclovir for congenital cytomegalovirus infection: systematic review and meta-analysis. Ultrasound Obstet Gynecol 2022, doi:10.1002/uog.26136.
  25. Faure-Bardon, V.; Fourgeaud, J.; Stirnemann, J.; Leruez-Ville, M.; Ville, Y. Secondary prevention of congenital cytomegalovirus infection with valacyclovir following maternal primary infection in early pregnancy. Ultrasound Obstet Gynecol 2021, 58, 576-581, doi:10.1002/uog.23685.
  26. Egloff, C.; Sibiude, J.; Vauloup-Fellous, C.; Benachi, A.; Bouthry, E.; Biquard, F.; Hawkins-Villarreal, A.; Houhou-Fidouh, N.; Mandelbrot, L.; Vivanti, A.J.; et al. New data on efficacy of valaciclovir in secondary prevention of maternal-fetal transmission of CMV. Ultrasound Obstet Gynecol 2022, doi:10.1002/uog.26039.

This manuscript is a resubmission of an earlier submission. The following is a list of the peer review reports and author responses from that submission.

Round 1

Reviewer 1 Report

This review paper summarized the strategies for the prevention of congenital CMV infection. This paper provided sufficient details with supporting evidence. Although the measures to prevent CMV infection are well reviewed in other papers, it is still worthwhile to update our knowledge and awareness from recent studies. However, I found that the authors did not cited references properly, some of them are listed below. Please check again your references thoroughly.

1.       Lines 88-91, “The transmission from child to adult…correlated with cCMV infection. Please provide citations for these claims.

2.       Lines 94, ”Recent studies..”  Please provide the references for these studies.  Also could you make it clear how high prevalence of CMV and prevalence of cCMV infections relate to important of secondary CMV infections.

3.       Line 111, “But a low level” should be “But a high level”

4.       Line 119, should be “their recent publications”

5.       Line 122: “The authors extended the investigation”. It is not clear what “the authors” refers to?

6.       Line 156-157: This sentence should be restructured. The “studies” should not have “positive attitude”

7.       Line 203-204: “…practice preventive hygiene behaviors.” again please provide the corresponding references for this and the following three sentences.

8.       Line 222: “Revello et al”. You cited some Rovello et al paper, please specify which one is referred here.

Reviewer 2 Report

Manuscript review:

Manuscript: Primary, Secondary, and Tertiary Prevention of Congenital Cytomegalovirus Infections.

Authors: Sartori, P, Egloff, C., Heini, N., et.al.

Journal:  Viruses

General comments: In this review, the authors have described different approaches that could be used to impact the incidence of congenital cytomegalovirus (cCMV) infection and thus decrease the public health importance of this perinatal infection. The lengthy text is reasonably comprehensive but in the opinion of this reviewer, requires the reader to have considerable background knowledge of the subject to interpret some of the more important points made by the authors. Moreover, there is a significant amount of discussion dedicated to patient education and attitudes about health education in the text. Although this subject is important, it seems out of place in the journal Viruses and would seem better received in a venue that focuses more on public health education. It is unlikely that most readers of Viruses will grasp the relevance of this information and therefore will likely have little interest in the content. Lastly, the text of this manuscript contains an extraordinarily large amount of non-standard English. In fact, some of the word choices and sentence structure often clouds the presumed content of the statement. It is not publishable in its current form. I can list a large number of examples but for instance, within the 1st page there are 4-5 or more examples which really detract from the content of the text. Some but not all specific queries are listed below:

1)    Lines 338-347. This is the extent of discussion of vaccines. In the field this is the topic that is driving most discussion and although it is not the only approach it deserves more detail than provided.

2)    Lines 391-395 are cryptic and a reader with knowledge of CMV infections in transplant patienst can fill in the details but other readers will have no idea of the specific point(s) this section is trying to convey.

3)    Lines 403-404 is wrong. Women with primary infections can have a + IgM and a +PCR.

4)    Lines 405-411 is completely uninterpretable and speculative at best.

5)    Line 425 is an example of non-standard English. “Controversying”. Similarly line 463 “virus then reproduces”

6)    Line 516 such statements if “debated” need discussion and references and not what appears to be a dogmatic claim.

7)    Line 542 is understood but it is not the “reference treatment” as it is not licensed in most countries in the northern hemisphere and this study has not been confirmed. Perhaps the best available data to date but a single study.

Overall recommendation: In its current form this review is not publishable for reasons noted above but particularly because the reader without sufficient background knowledge will not be provided with an objective review of this subject nor identifiable areas that require further study. The English usage only adds to the issues surrounding the content.

Reviewer 3 Report

The authors provide a comprehensive review of key issues surrounding prevention of CMV infection and CMV disease in the fetus, as it relates to CMV awareness and behavioral and pharmacologic interventions. 

There are references that may be considered to be included, but not critical to be included, as of course it is impossible to include all references:

Baer H et al J Clin Virol 2014  60(3): 222 - this is a survey study on USA medical students CMV awareness

Dedhia et al.  Amer J Audiol Mar 2021 - survey study of USA audiologists and S/L therapists CMV awareness

Almishaal, et al. PLOS ONE Sept 2022- just published survey study on Saudi Arabian women CMV knowledge